# Large-scale online assessment uncovers a distinct Multiple Sclerosis subtype with selective cognitive impairment

Annalaura Lerede [1,2] ✉, Alexandra Moura[2], Valentina Giunchiglia[1,2,3], Elisa Carta[4], William Trender [2], Katherine Tuite-Dalton[5], James Witts [5], Elaine Craig[5], Sarah Knowles [5], Jeff Rodgers[5], Eleonora Cocco [6], Peter J. Hellyer[1], Rod Middleton [5], Richard Nicholas [2,5,7] ✉ & Adam Hampshire[1,2,7]

Cognitive impairments in Multiple Sclerosis (MS) are prevalent and disabling yet often unaddressed. Here, we optimised automated online assessment technology for people with MS and used it to characterise their cognitive deficits in greater detail and at a larger population scale than previously possible. The study involved 4526 UK MS Register members over three stages. Stage 1 evaluated 22 online cognitive tasks and established their feasibility. Based on MS discriminability a 12-task battery was selected. Stage 2 validated the resulting battery at scale, while Stage 3 compared it to a standard neuropsychological assessment. Clustering analysis identified a prevalent MS subtype exhibiting significant cognitive deficits with minimal motor impairment. Disability in this group is currently unrecognised and untreated. These findings underscore the importance of cognitive assessment in MS, the feasibility of integrating online tools into patient registries, and the potential of such large-scale data to derive insights into symptom heterogeneity.

Cognitive impairment is prevalent in Multiple Sclerosis (MS), affecting 40%–70% of patients[1,2]. It can occur at any disease stage, including at the extremes (e.g., clinically isolated syndrome[3], paediatric-onset[4,5] and late-onset[6]), and across all subtypes[7], even in the absence of other symptoms[8]. Its impact on quality of life and employment is often greater than that of physical impairment[9,10]. Cognitive deficits in MS most commonly affect information processing speed, working memory, sustained attention, and memory encoding, but can also involve executive functions and visuospatial abilities, albeit less frequently[1,2,11–13].

Despite its importance, cognitive impairment is rarely considered as an outcome in clinical trials, is not routinely assessed in clinical practice, and is absent from current clinical phenotype definitions[7,14,15]. This is partly due to the limitations of standard neuropsychological assessments, which are typically conducted in person, require a trained examiner, and are either lengthy or, in the case of shorter assessments, lack comprehensiveness[16,17]. As a result, large-scale and longitudinal evaluations remain impractical, and systematic data on MS-related cognitive impairment are scarce—hindering progress in understanding its mechanisms, impact, underlying aetiology, associations with disease-related factors, and the efficacy of available treatments on cognition[14].

Online cognitive assessment tools offer a promising solution to address these knowledge gaps. They are accessible, cost-effective, and

[1]Centre for Neuroimaging Sciences, Institute of Psychiatry, Psychology & Neuroscience, King's College London, London, UK. [2]Department of Brain Sciences, Imperial College London, London, UK. [3]Department of Biomedical Informatics, Harvard University, Boston, MA, USA. [4]Multiple Sclerosis Centre, ASL Cagliari, Department of Biomedical Sciences, University of Cagliari, Cagliari, Italy. [5]Population Data Science, Swansea University Medical School, Swansea, UK. [6]Multiple Sclerosis Centre, ASL Cagliari, Department of Medical Sciences and Public Health, University of Cagliari, Cagliari, Italy. [7]These authors contributed equally: Richard Nicholas, Adam Hampshire. ✉e-mail: annalaura.lerede@kcl.ac.uk; r.nicholas@imperial.ac.uk

efficient[18,19]. They can be performed remotely on commonly used personal devices without specialised personnel or equipment[20,21], enabling wide-reaching participation–particularly among individuals with mobility limitations[19]. These tools can capture a broad range of cognitive domains, enabling large-scale, comprehensive, and longitudinal data collection[22–24]. Standardised administration and automatic scoring reduce the potential for biases and human errors[25,26], while the collection of detailed time courses of task performance offers more sensitive and dynamic measures of cognitive function[22,27]. This also allows for the extraction of diverse behavioural metrics, offering deeper insights into cognitive processes[28,29]. Despite their potential, their feasibility, validity, and clinical utility in MS still require comprehensive evaluation. Existing studies have typically relied on small sample sizes and assessed a limited range of tasks, primarily focused on information processing speed[30,31].

The presented observational study was a 3-stage collaboration with the UK MS Register (UKMSR) to develop an online cognitive assessment battery tailored to measure and monitor at scale the aspects of cognition most impaired in people with MS (pwMS). In Stage 1, we evaluated the feasibility of using an established platform– Cognitron[22–24,27,29]–for deploying cognitive tasks remotely and under unsupervised conditions for pwMS. This was the first large-scale deployment of Cognitron for pwMS. From this, we curated a subset of tasks to form the Cognitron-MS (C-MS) battery–designed to comprehensively measure cognitive function while maximising the discriminability of pwMS from controls. In Stage 2, we validated the C-MS battery in an independent MS sample, and in Stage 3, we cross-compared it with a standard in-person neuropsychological assessment, to evaluate its comparative validity and clinical utility. Finally, we applied data-driven clustering to cognitive performance data from the C-MS battery and motor disability data from patient-reported outcomes (PROs) to identify and characterise distinct symptom-based phenotypes, including a common subtype with selective cognitive impairment. In doing so, we aimed to advance individualised symptom profiling in MS by incorporating detailed cognitive characterisation, which is often underrepresented in large-scale MS studies due to the lack of appropriate and scalable assessment tools.

## Results

### Online cognitive assessment is feasible in people with MS

Among 19,188 UKMSR members emailed, 3066 engaged with the assessment in Stage 1 and 2696 in Stage 2 (Fig. 1B, N by task in Tables S1 and S2, sociodemographic characteristics in Tables S3 and S4). For comparison, between 3198 (Stage 1) and 3356 (Stage 2) UKMSR users were considered active in the same timeframe based on the completion of all the core PROs the last two times of asking. Stage 1 participants did not significantly differ from the overall UKMSR population in terms of age and disease duration (all p-values and summary statistics in Table S5) but had a higher prevalence of females (77.4% compared to 74.2%). 41.3% of Stage 1 participants returned to engage with Stage 2; returners were older on average (returning: 56.5 years; non-returning: 52.9 years) and had a higher prevalence of progressive individuals (returning: benign–BN = 2.4%, relapsing-remitting–RR = 49.4%, secondary progressive–SP = 32.3%, primary progressive–PP = 15.9%; non-returning: BN = 2.1%, RR = 57%, SP = 28.3%, PP = 12.6%). No differences were observed in sex, disease duration, or severity. 17% of participants in Stage 1 and 31.6% in Stage 2 did not complete any of the available PROs during the collection windows closest to the cognitive assessments.

78.4% of participants in Stage 1 and 81.5% in Stage 2 completed the assessment (Fig. 1B), with 73.3% and 70.4% respectively having done so in a single session (i.e., without taking breaks; mean duration: Stage 1 = 42 min, Stage 2 = 46 min; median duration: Stage 1 = 39 min, Stage 2 = 42 min). Completion of the assessment did not depend on age,

disease subtype, or duration, but was modestly associated with sex and severity measures (all p-values and summary statistics in Table S5). At Stage 2 only, completers had a higher proportion of RR individuals compared to non-completers.

### Tailoring a battery of online cognitive tasks for MS

To inform the selection of an MS-optimised online cognitive battery, we first assessed which domains showed the most prominent impairments in pwMS. Significant deficits in cognitive performance compared to controls were observed in Stage 1 participants (N = 3048, N by task in Table S2) across most evaluated tasks (Figs. 1A and S20), with effect sizes–quantified as median Deviation from Expected (mDfE) scores across task performance metrics– ranging from small to large (Fig. 2A). These group differences remained robust across alternative outlier removal strategies (Figs. S1 and S2) and after re-estimating them following propensity score matching of pwMS and controls on sociodemographic covariates (Fig. S3). Sustained attention, executive functions, visuospatial problem solving, working memory, and information processing speed (IPS) were the cognitive domains measured by the tasks showing the largest effect sizes. Notably, Switching Stroop and Trail Making cost metrics confirmed executive function deficits, even when the contribution of visuomotor impairment was subtracted out.

Next, we confirmed the specificity of the observed deficits to cognition. Accuracy and cost metrics showed largely non-significant or negligibly small correlations with PROs related to physical disability, anxiety, depression, fatigue, and quality of life (Fig. 2C top), supporting their specificity to cognition. Conversely, response time metrics correlated moderately with PROs measuring physical disability (MS Impact Scale–MSIS-29 motor, MS Walking Scale–MSWS-12, and web Expanded Disability Status Scale–webEDSS), indicating that motor impairment may partially influence response speed in some of the tasks (Fig. 2C bottom). As a result, accuracy was selected as the primary performance metric for most tasks, except those explicitly designed to measure response time (SRT, CRT, Motor Control, and Trail Making).

Finally, to minimise redundancy when selecting tasks for the MS-optimised battery, we identified their underlying cognitive structure. Factor analysis of primary performance metrics across tasks identified a 6-factor solution based on eigenvalues > 1 (Kaiser's criterion; scree plot in Fig. S4). These factors cumulatively explained 28.6% of the variance (Table S6) and were interpreted as IPS, visuospatial problem solving, working memory, verbal abilities, memory, and attention. The validity of the factor solution (Fig. S5) and its stability at the available sample size (Fig. S6) were confirmed via iterative split-half cross-validation (details in Supplementary Note S1).

One to three tasks with high loadings onto each of the six factors were selected to form the C-MS battery (Fig. 2B), prioritising tasks with strong MS discriminability (Fig. 2A) and low sensitivity to the device type (Fig. S7). The selected 12 tasks were Word Memory Immediate, Motor Control, Manipulations 2D, Card Pairs, Switching Stroop, Blocks, Verbal Analogies, Word Definitions, SRT, Trail Making, Word Memory Delayed, and Spotter. Words Memory Immediate and Delayed were selected despite lower MS discriminability, as they assess the memory domain and are sensitive to impairments in other conditions relevant for cross-cohort comparisons. SRT and Motor Control were retained despite their higher device sensitivity, as they capture the information processing and visuomotor domains. Although no factor directly captured executive functions, Switching Stroop and Trail Making allow measurement of deficits in this domain via their cost metrics.

The resulting 12-task C-MS battery (Figs. 1A and 2D) was administered end-to-end in Stage 2, along with two additional tasks–

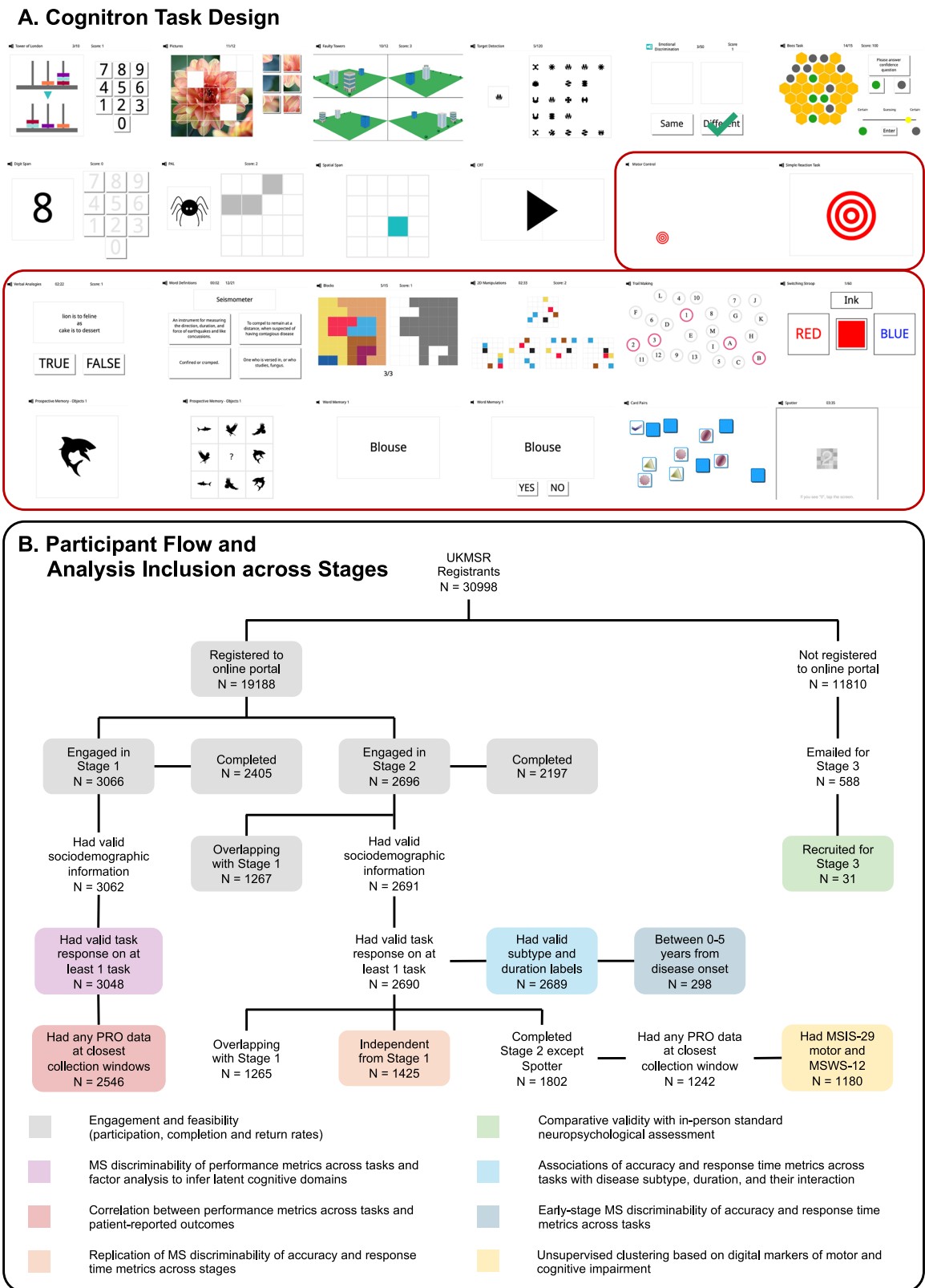

**Fig. 1 | Cognitron task design and participant flow. A** Overview of the 24 tasks from the Cognitron platform employed in the study. Twenty-two tasks were evaluated in Stage 1 using a random task sampling approach. Two additional tasks were introduced in Stage 2. Tasks circled in red were selected in Stage 1 to form the C-MS battery used in Stages 2 and 3. **B** Recruitment steps from the wider UK Multiple Sclerosis (MS) Register cohort and analysis steps. The total number of people with MS (pwMS) in each group is reported. The number of pwMS available for each task varies due to random task sampling and/or assessment completeness and is detailed in Tables S1 and S2. The groups used for each analysis are indicated by boxes with different colours.

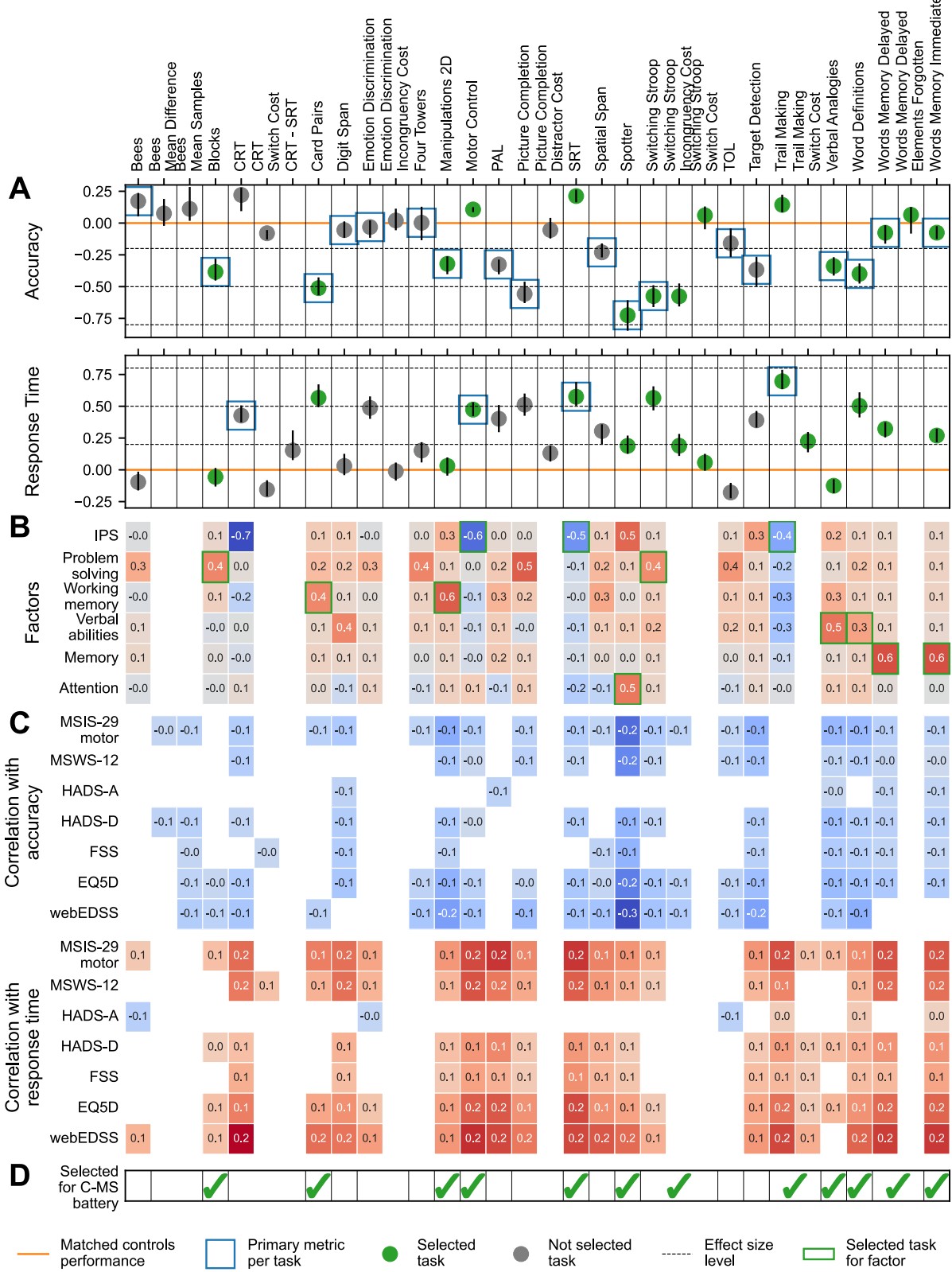

Objects Memory Immediate and Delayed—which demonstrated high sensitivity to subtle memory deficits in other populations[22,23,32].

### Validation of the C-MS battery: replication of MS discriminability across stages and comparative validity

To confirm our task selection for the C-MS battery, we replicated findings in an independent cohort and evaluated the addition of the two objects memory tasks. A total of 1425 participants from Stage 2 engaged with the assessment for the first-time, providing an independent validation sample (N by task in Table S2). When their pattern of MS discriminability across tasks was compared to that from Stage 1, the correlations were large (accuracy: $r = 0.93$, $p < 0.0001$; response time: $r = 0.88$, $p = 0.0004$; Fig. 3A). Moreover, the two additional objects memory tasks demonstrated greater MS discriminability

**Fig. 2 | Multiple Sclerosis (MS) discriminability across task performance metrics and task selection for the C-MS battery. A** MS discriminability in terms of accuracy, cost in accuracy (top), response time, and cost in response time (bottom) across tasks for Stage 1 participants (N = 3048; task-specific Ns in Table S1). MS discriminability is defined as the median deviation from expected (DfE) score across participants with 95% confidence intervals for each performance metric. Tasks selected for inclusion in the C-MS battery are reported in green. The primary performance metric for each task is indicated by a blue square. The orange line represents the control median. **B** Heatmap of factor loadings across primary performance metrics. Tasks selected for each factor are indicated by green squares. **C** Heatmap of correlations between task performance metrics (top: accuracy and cost in accuracy; bottom: response time and cost in response time) and patient-reported outcomes (PROs) including MSIS-29 motor, MSWS-12, HADS-A, HADS-D, FSS, EQ5D and webEDSS. Only statistically significant (p ≤ 0.05) correlations are shown. **D** Grid indicating with a green check mark whether a task was selected for the C-MS battery. Source data are provided as a Source Data file.

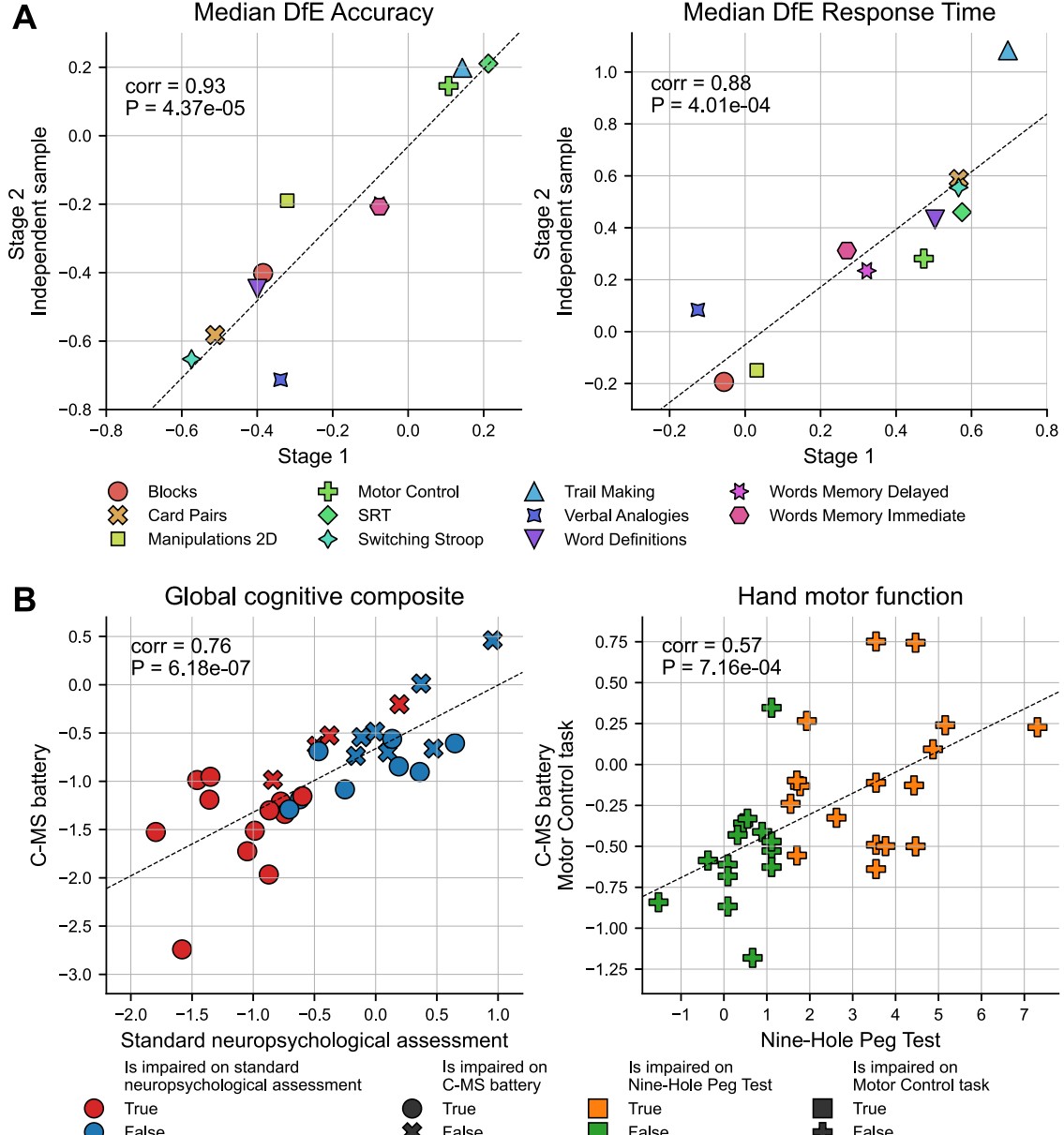

**Fig. 3 | Replication of Multiple Sclerosis (MS) discriminability across stages and comparative validity with standard neuropsychological assessment. A** Scatter plots showing MS discriminability in terms of accuracy (left) and response time (right) across tasks for Stage 1 (x-axis, N = 3048) and the independent sample from Stage 2 (y-axis, N = 1425). MS discriminability is defined as the median deviation from expected (DfE) score across participants for each performance metric and is evaluated separately for each stage. Two-tailed Pearson correlation coefficients and corresponding p-values between values at Stage 1 and Stage 2 for accuracy and response time are reported on the plots. **B** Scatter plots showing global composites of cognitive performance (left) and hand motor function (right) across Stage 3 participants (N = 31), as measured by the standard neuropsychological assessment (x-axis) and the C-MS battery (y-axis). Global cognitive composites were derived by averaging standardised performance scores across tasks within each battery. Nine-Hole Peg Test scores correspond to the less impaired hand. Two-tailed Pearson correlation coefficients and corresponding p-values between global cognitive composites and hand motor function scores from the standard and online assessments are reported on the plots. Source data are provided as a Source Data file.

(Objects Memory Immediate: mDfE accuracy = −0.52, mDfE RT = 0.43; Objects Memory Delayed: mDfE accuracy = −0.54, mDfE RT = 0.44; Fig. S20) compared to the words memory tasks already present in Stage 1 (Words Memory Immediate: mDfE accuracy = −0.2, mDfE RT = 0.31; Words Memory Delayed: mDfE accuracy = −0.16, mDfE RT = 0.29).

To evaluate comparative validity against a standard neuropsychological assessment, we analysed data from the 31 pwMS who participated in Stage 3 (sociodemographic characteristics in Table S7). Among them, 16 were identified as cognitively impaired using the standard neuropsychological assessment. The C-MS battery successfully detected cognitive impairment in 75% of them, classifying 20 participants overall as impaired. Furthermore, global cognitive composites derived separately for each battery were strongly correlated ($r = 0.76$, $p < 0.0001$; Fig. 3B left). Performance on the less impaired hand of the Nine-Hole Peg Test positively correlated with the Motor Control task ($r = 0.57$, $p = 0.0007$; Fig. 3B right), included in the C-MS battery to gauge visuomotor function. Additionally, it also correlated with Trail Making ($r = 0.4$, $p = 0.0243$) and Manipulations 2D ($r = −0.37$, $p = 0.0427$), but not with the other tasks (Table S8). Notably, none of the participants were classified as impaired on the Motor Control task, whereas 17 were impaired on the Nine-Hole Peg Test.

## Validation of the C-MS battery: associations of cognitive impairment with disease-related factors

To further validate the C-MS battery, we examined its associations with established disease-related factors, leveraging our large Stage 2 cohort to replicate known trends in MS cognitive impairment.

For Stage 2 participants who completed the assessment ($N = 1802$), the overall prevalence of cognitive impairment was evaluated using three different thresholds, resulting in 72.1%, 29.4% and 14.6% of the participants being classified as impaired using, respectively a liberal, fair, and conservative criterion. Irrespective of the criterion applied, cognitively impaired participants were significantly older and had longer disease durations. Additionally, the impaired groups had a lower proportion of females and RR individuals compared to the non-impaired groups (Table S9).

In the broader Stage 2 cohort ($N = 2689$, $N$ by task in Table S2), performance metrics showed mostly non-significant associations with disease duration when taking into account subtype, except for response times in Trail Making, Blocks, and Objects Memory Immediate and Delayed, with RR individuals generally responding faster than progressive individuals (Figs. 4A, S24 and 25). These differences were particularly pronounced for Trail Making. Conversely, disease subtype was significantly associated with response times across all tasks and with accuracy in Card Pairs, Manipulations 2D, Motor Control, Objects Memory Immediate, Switching Stroop, and Verbal Analogies (Figs. 4A, S22 and 23), with RR individuals generally outperforming progressive individuals. Pairwise comparisons are reported in Figs. S8 and S9.

Focusing on participants within 5 years of disease onset ($N = 298$, $N$ by task Table in S10), most tasks discriminated between early-stage pwMS and controls either in terms of accuracy, response time, or both (Fig. 4B). Trail Making, with a large effect size, followed by Verbal Analogies, Object Memory Immediate and Delayed, with medium-large effect sizes, were the tasks showing the largest differences, indicating early deficits in IPS, verbal reasoning, and objects memory when comparing to the control population.

## Unsupervised clustering of people with MS based on digital markers of motor and cognitive impairment

We integrated the C-MS battery with established motor disability measures—MSIS-29 motor and MSWS-12—to derive symptom-based, data-driven MS subtypes.

Based on the clustering stability analysis (see Supplementary Note S2 and Figs. S10 and S12), a 4-cluster $K$-Means solution emerged as optimal for grouping Stage 2 participants with available motor PROs ($N = 1180$) using selected motor and cognitive features (Fig. 5A). The four clusters comprised 26.0%, 34.2%, 25.5% and 14.3% of pwMS, respectively. Two clusters (26% and 25.5%) showed minimal motor impairment (median MSIS-29 motor = 20/25, median MSWS-12 = 18.8/22.9), while the other two (34.2% and 14.3%) exhibited severe motor impairment (median MSIS-29 motor = 65/63, median MSWS-12 = 100/85.4), with the former including more individuals unable to walk (54.8% vs. 47.3%). Cluster membership was significantly associated with both MSIS-29 motor and MSWS-12 scores (all $p < 0.0001$).

The clusters with minimal motor impairment differed in the presence of cognitive deficits. The 26.0% cluster showed medium-to-large deficits on Objects Memory Immediate and Delayed, Card Pairs, Switching Stroop, Verbal Analogies, Word Definitions, and Trail Making, indicating widespread cognitive problems (Figs. 5B and S26). With 44.6% of pwMS in this cluster classified as cognitively impaired (compared to 35.6% overall), it was labelled Minimal Motor + Moderate Cognitive. The 25.5% cluster exhibited relatively preserved cognitive function, except for slower Trail Making (mDfE = 0.34), with only 1% cognitively impaired, and was thus labelled Minimal Motor + No Cognitive.

Conversely, the clusters with severe motor impairment differed in cognitive deficit severity. The 34.2% cluster exhibited medium-to-large deficits on Switching Stroop, SRT, and Trail Making, and small-to-medium deficits across most other tasks, indicating widespread but milder impairments (Figs. 5B and S26). With 28.3% classified as cognitively impaired, it was labelled Severe Motor + Mild Cognitive. Lastly, the 14.3% cluster showed large deficits across all tasks, consistent with advanced cognitive impairment. With 98.2% of pwMS classified as cognitively impaired, it was labelled Severe Motor + Severe Cognitive.

Significant associations were found between cluster membership and cognitive features (all $p < 0.0001$). Post-hoc pairwise comparisons among clusters for both motor and cognitive features are reported in Fig. S13.

## Insights using the C-MS battery: clusters characterisation with unseen variables

Finally, we further characterised the derived clusters by examining trends in sociodemographic variables not included in the clustering. Cluster membership was associated with age, sex, education level, disease duration, and subtype (Fig. 5A and Table S11). The Minimal Motor + Moderate Cognitive cluster had the youngest average age, while the Severe Motor + Mild Cognitive cluster the oldest, though their mean age difference was only 6.7 years; all age groups were present in the younger cluster. Sex distribution was similar across clusters except for the Severe Motor + Severe Cognitive cluster, which had the lowest proportion of females (up to 17.9% less than others). Notably, university degree prevalence was highest in clusters with minimal motor impairment and lowest in the Severe Motor + Severe Cognitive cluster. Mean disease duration was shorter in minimal motor impairment clusters and longer in severe clusters, differing by up to 5.3 years on average; however, all durations were present in every cluster. Finally, the proportion of RR individuals decreased, and of progressive individuals (SPMS and PPMS) increased with motor impairment severity, with the Severe Motor + Mild Cognitive showing the highest prevalence of progressive individuals. Nonetheless, all MS subtypes were present in every cluster.

Examining PROs excluded from clustering in pwMS for whom these data were available, significant associations were found for Hospital Anxiety and Depression Scale (HADS-D and HADS-A), Fatigue Severity Scale (FSS), EQ5D and webEDSS (Fig. 5B). Median depression and anxiety scores increased with motor impairment severity, as did

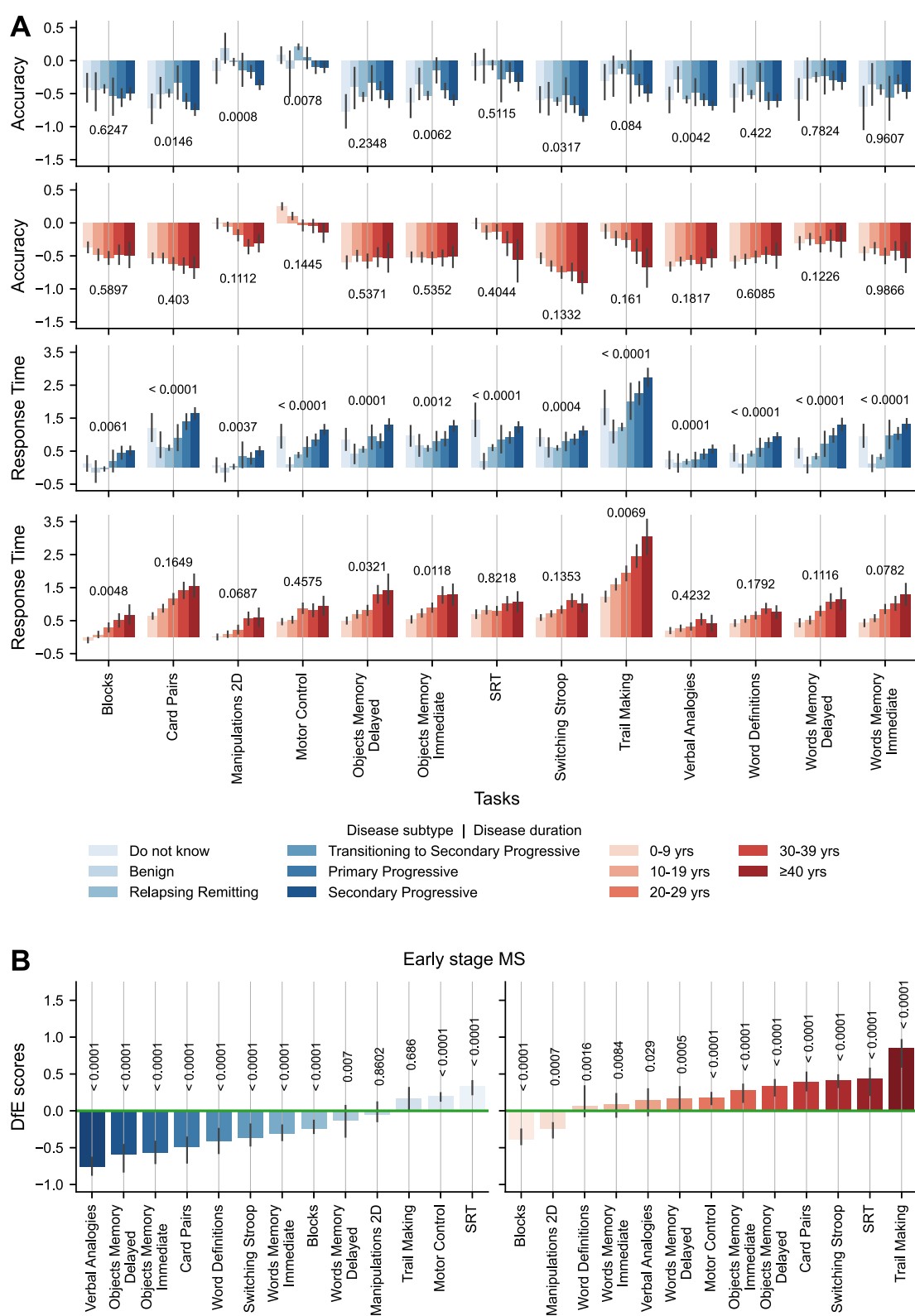

fatigue, reduced quality of life, and disease severity, measured by FSS, EQ5D and webEDSS, respectively (Figs. 5B and S26; pairwise comparisons in Fig. S14).

## Discussion

Our results demonstrate the feasibility of online cognitive assessments in pwMS through the collection of an unprecedented scale of objective task performance data from a representative sample over a relatively short timeframe[33,34]. This was made possible by combining the efficiency and accessibility of online cognitive assessments[18,19] with the reach of the UKMSR[34,35].

A total of 4495 pwMS participated across Stages 1 and 2–making this, to our knowledge, the largest study on cognition in MS to date. Their data enabled the identification of a concise battery of online

**Fig. 4 | Associations of cognitive performance with disease subtype and duration, and task discriminability of early-stage people with Multiple Sclerosis (pwMS) and controls. A** Mean deviation from expected (DfE) scores with 95% confidence intervals for accuracy (top) and response time (bottom) across all tasks, stratified by disease subtype (shades of blue) and disease durations (shades of red), for participants who reported both subtype and onset year in the questionnaire (*N* = 2689; task-specific Ns in Table S2). Two-way permutational ANOVA was performed separately for each performance metric to test for main and interaction effects of subtype and disease duration. **B** Median DfE scores with 95% confidence intervals for accuracy (left) and response time (right) across tasks for participants with early-stage MS (between 0 and 5 years post disease onset, *N* = 298; task-specific Ns in Table S10). Two-tailed Wilcoxon signed-rank tests against zero were used to assess statistical significance. Resulting *p*-values are reported in the figure. *P*-values < 0.0001 are reported as <0.0001 due to precision limits based on 10,000 permutations. Source data are provided as a Source Data file.

tasks that assesses detailed aspects of cognition whilst being specifically tailored for detecting impairments in MS and deployable in a home setting. Participation rates were notable: 16% of total UKMSR users in Stage 1 and 14% in Stage 2, achieved over 7 and 2 months, respectively. These compared favourably to the rates normally achieved by the UKMSR, with 17% of total UKMSR users being considered as active around the timeframe of the study.

Completion rates increased from 78.4% in Stage 1 to 81.5% in Stage 2, indicating successful battery optimisation. These rates were comparable to other online studies using Cognitron, including in an older adult birth cohort[36] and a recontacted UK epidemiological sample[23]. The absence—or minimal impact—of age, disease subtype, duration, and severity on assessment completion supports its feasibility in pwMS.

Another positive outcome was that 41.3% of Stage 1 participants returned for Stage 2, despite the cognitive battery being under development, more demanding than PROs, and not yet returning individual results. Returnees were on average older and had a higher prevalence of progressive MS, suggesting that those at greater risk of cognitive impairment[37–39] were not less likely to re-engage. While this return rate was lower than that observed for PROs (e.g., MSIS-29: 71.1%), it is expected to improve as the battery becomes more optimised, widely validated, and familiar to participants.

The C-MS battery demonstrated strong discriminative validity, with pwMS performing worse than controls across all tasks, even after adjusting for detailed sociodemographic factors and device variability. It also showed good comparative validity with a standard neuropsychological assessment and increased precision. The cognitive domains measured by the tasks showing the highest MS discriminability converged with those found to be most affected in MS literature, further validating the assessment[1,2,11]. Additionally, online tasks offered more granular measures of these domains, capturing trial-by-trial performance data that can be summarised in a number of metrics or analysed in their entirety[28,40]. The MS discriminability of the selected tasks replicated in an independent sample, confirming their selection for the C-MS battery and their general validity in consistently capturing deficits at the population level.

Using the collected data, we investigated associations between cognitive function and disease-related factors with greater precision and statistical power than has previously been possible for MS. The prevalence of global cognitive impairment in the complete sample from Stage 2 aligned with values reported in the literature[41] and supported the notion that impairment is more prevalent in older pwMS, in those with longer disease duration, and in progressive subtypes[37]. However, variability in prevalence rates across different thresholds (14.6%–72.1%) highlighted the need to measure cognition as a continuous function. Indeed, the accumulation of cognitive impairment is a continuous phenomenon, and establishing a threshold inevitably results in overlooking changes occurring at either side of the threshold[42]. To address this, it is crucial to assess cognition at the onset and monitor changes over time with precision instruments. While this level of scale, detail, and repetition is unfeasible with standard neuropsychological assessments, it is trivial with well-designed online cognitive tasks.

Aligning with existing literature, we found more severe impairments in progressive subtypes, particularly in SPMS[38,43], which occurs later in the disease course, but also widespread deficits early in the disease[44]. Notably, when analysing cognitive accuracy scores, associations with disease duration were accounted for by the inclusion of subtype labels. This was also the case for all but four tasks when analysing response time scores. The literature is inconclusive on the relationship between cognitive deficits and disease duration[1,2,45]. One possibility is that cognitive change primarily occurs early on and around transition points, e.g., disease onset and phenoconversion. Cross-sectional designs are suboptimal for determining if this is the case. However, periodic online cognitive assessments will produce the longitudinal data needed to address this question in detail.

The absence of a distinct cognitive fingerprint for specific MS subtypes and stages, reflects the underlying disease's heterogeneity, driven by individual patterns of neuroinflammation and neurodegeneration[46,47]. To better account for this variability, we explored a data-driven clustering approach using digital cognitive and motor markers, aiming to identify clinically meaningful subgroups beyond conventional classifications.

Through this approach, we uncovered a prevalent group—over a quarter of our cohort—with substantial cognitive deficits in the absence of notable physical disability (Minimal Motor + Moderate Cognitive cluster). These individuals spanned all ages and disease durations, though they were on average younger and earlier in their disease course, suggesting that cognitive deficits can emerge early and in isolation, but also persist selectively over time.

Crucially, this selective cognitive MS subtype likely remains undetected by standard motor-focused clinical evaluations, leading to underestimation of disability—both at onset and throughout the disease course—and potentially delayed or inadequate treatment. Supporting this, webEDSS scores were lowest in this group, and their correlations with cognitive performance metrics were generally weak. Our findings align with previous reports showing that 45–47% of individuals classified as having Benign MS exhibit cognitive impairment[15,48], and that EDSS can underestimate disability when cognitive function is not evaluated[49].

Moreover, the cognitive deficits observed in this cluster encompassed domains beyond information processing speed, including memory, cognitive control, associative working memory, and verbal reasoning. While information processing speed deficits remain the most frequently reported[13,50,51] and prioritised during brief assessments[16], our findings suggest that focusing solely on this domain would likely overlook substantial cognitive impairment in this cluster.

Taken together, these findings raise the possibility that this selective cognitive subtype reflects distinct patterns of underlying neuropathology, rather than simply representing an early-stage of more global deterioration. One plausible mechanism is the known early cortical damage seen in MS, particularly in the superior frontal cortex, which has been associated with cognitive changes in both clinically isolated syndrome and early MS[52]. It is increasingly evident that such cortical damage can occur before clinical presentation—thus in isolation and without impacting standard mobility measures[53,54]. Further biomarker and neuroimaging studies are warranted to clarify the mechanisms underlying this phenotype and to determine whether it represents a stable trajectory or a transitional phase in MS progression.

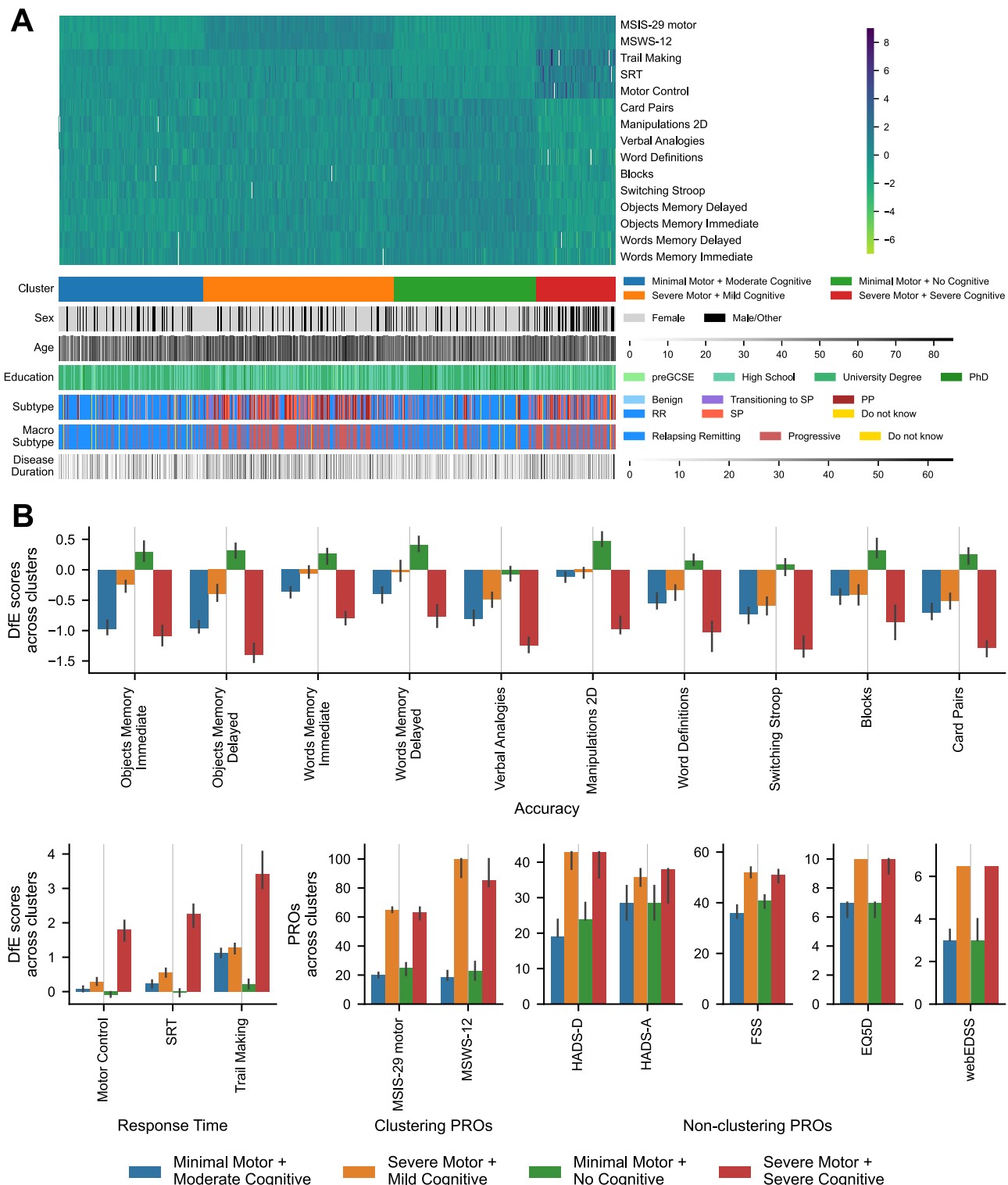

**Fig. 5 | Unsupervised clustering of people with Multiple Sclerosis (pwMS) based on digital markers of motor and cognitive impairment and characterisation of resulting clusters. A** Heatmap (top) showing standardised motor and cognitive scores for all participants (*N* = 1180), sorted by cluster membership. Clusters were derived using *K*-Means clustering applied to 13 cognitive features (one per task) and 2 motor features (one per patient-reported outcome−PRO). A 4-cluster solution was selected as the most valid and stable after comparing different clustering methods and cluster numbers. Bottom heatmaps display the distribution of participant characteristics−sex, age, education, MS subtype and disease duration−across clusters. **B** Median scores with 95% confidence intervals for each cluster (Minimal Motor + Moderate Cognitive: *N* = 307; Severe Motor + Mild Cognitive: N = 403; Minimal Motor + No Cognitive: *N* = 301; Severe Motor + Severe Cognitive: *N* = 169) across clustering variables (motor and cognitive features) and additional clinical variables (HADS-D, HADS-A, FSS, EQ5D and webEDSS). Source data are provided as a Source Data file.

Another interesting finding from the symptom-based clustering was the higher prevalence of males in the Severe Motor + Severe Cognitive cluster, resulting in a distinctive male-to-female ratio compared to the other clusters. This difference could not be attributed to an increased prevalence of PP individuals within this cluster, as this prevalence was greater in the Severe Motor + Mild Cognitive cluster. This suggests a potential sex-related susceptibility to more severe cognitive impairment, a result that aligns with some past studies[49,55–57].

Overall, online cognitive assessment enables widespread collection of cognitive markers of disability, offering additional insights into this often-neglected aspect of the disease. While not a replacement for conventional clinical phenotyping, the presented data-driven classification offers a complementary perspective—focusing on specific impairments as opposed to clinical manifestations arising from postulate disease mechanisms. This approach could help guide personalised treatment and identify individuals with unrecognised cognitive impairment.

Although the use of an established digital registry enabled the cognitive assessment of a large and inclusive MS sample, a limitation is that individuals unfamiliar with or less willing to engage in online technologies may not have taken part in Stages 1 and 2, potentially introducing selection biases. More broadly, digital poverty remains a known challenge for such technologies. As per work in other populations[36], future studies should characterise the relationship between engagement in online assessments and demographics in MS, and explore ways to mitigate these barriers, but also consider applying the same technology under supervised conditions for the minority of pwMS where this is necessary[22].

Another key consideration is the potential influence of visuomotor impairments on cognitive task performance in an online setting. While accuracy and cost measures were designed to be robust to such impairments, response times were more susceptible, as shown by correlations with motor PROs. To some degree, these correlations may reflect the co-development of motor and cognitive impairment. To address this, we intend to further develop our modelling framework to disentangle visuomotor and cognitive components of task performance[28,40]. Notably, such a method cannot readily be developed for standard neuropsychological tests as it requires rich trial-by-trial data, available only for computerised assessments. Nonetheless, most of the primary performance metrics used in this study appeared largely robust against motor deficits and were accessible for most pwMS, including those with some motor impairment. In fact, in Stage 3, all participants completed the tasks with motor performance within the normal range, despite the presence of motor impairments within this MS sample.

In summary, we validated the application of fully automated large-scale online cognitive assessment for pwMS. By leveraging the resulting large population data, we identified a prevalent selective cognitive subtype that currently goes undetected and untreated but that can be effectively addressed using the technology applied in this study.

## Methods
Ethical approval for this study was covered under the UKMSR ethics framework. The UKMSR received ethical approval from the South-West Central Bristol National Research Ethics Service, initially as 16/SW/0194, and subsequently 21/SW/0085. All participants provided informed consent to be involved in the study.

### Study design
This observational study was conducted in collaboration with the UKMSR, a validated, UK-wide, online patient register launched in 2011 and involving nearly 20,000 pwMS[33,58]. The UKMSR includes independently verified treatments and EDSS outcomes from NHS centres as well as PROs collected regularly using validated online questionnaires (since September 2018, 6-monthly, over a 28-day window).

This study comprised three stages. Stage 1 was conducted between October 2021 and May 2022. Using a random task sampling approach, we evaluated a superset of online cognitive tasks from the Cognitron platform (Fig. 1A) in a UKMSR cohort to assess their feasibility and identify a concise MS-relevant battery (C-MS battery). Stage 2 was conducted between November 2022 and January 2023, administering the C-MS battery derived in Stage 1 to a partially overlapping UKMSR cohort to evaluate its validity. We assessed validity in two ways: (i) replicating Stage 1 results in an independent sample of pwMS, and (ii) testing associations between cognitive performance and disease-related factors against established literature. In Stage 3, conducted between January and March 2024, we assessed the comparative validity of the C-MS battery against a standard neuropsychological assessment in an independent MS cohort, with their order of administration randomised. Finally, leveraging the large-scale performance data collected in Stage 2, we demonstrated the utility of the C-MS battery by deriving data-driven insights into symptom heterogeneity in pwMS.

For Stages 1 and 2, invitations were emailed to all valid UKMSR registrants ($N = 19,188$ with year of birth, sex, subtype at diagnosis, and email available) to participate in the study. Participants were asked to complete a sociodemographic questionnaire (questionnaire transcript in Table S12) and an online computerised cognitive assessment using their preferred device (desktop computer, laptop, tablet, or smartphone). The Stage 2 questionnaire additionally asked for the year of symptoms onset and current disease subtype (possible options included benign–BN, relapsing-remitting–RR, secondary progressive–SP, transitioning to secondary progressive–TrnSP, primary progressive–PP, and do not know–DK). Two reminder emails were sent during each stage to participants who had not started or not completed the assessment. Data collected were linked to PROs available for participants in the UKMSR, including the MS Impact Scale version 2 (MSIS-29)[59,60], MS Walking Scale (MSWS-12)[61], Hospital Anxiety and Depression Scale (HADS)[62,63], EuroQol Five Dimensions of Life 3-level version (EQ5D)[64], Fatigue Severity Scale (FSS)[65], and web Expanded Disability Status Scale (webEDSS)[66]. Linkage with the PROs was performed only for participants who had completed them during the collection windows closest to the cognitive assessment. For Stage 1, which was more spread out, the window was either before (Autumn 2021) or after (Spring 2022) the assessment, based on the temporally closest available PRO dataset. For Stage 2, the window before the assessment (Autumn 2022) was used for all participants, as it was closer in time than the following one.

For Stage 3, patients from the Hammersmith Hospital site who had clinically consented to be part of the UKMSR but had not yet registered on the online portal ($N = 588$) were invited via email to participate in the study. Responders ($N = 31$) were scheduled for an in-person appointment to perform both the C-MS battery on a tablet and the standard neuropsychological assessment with an examiner. Additionally, they were administered the Nine-Hole Peg Test (9HPT)[67] to gauge their hand motor function and determine if it might impede engagement with online cognitive tasks. Figure 1B reports the number of pwMS who participated in each stage and those retained for the different analyses.

### Cognitive assessment
The online cognitive assessments were hosted by the Cognitron website and embedded within the UKMSR website, which cohort members were more familiar with. Cognitron offers a comprehensive suite of tasks designed to measure different cognitive domains while being straightforward and user-friendly. Some tasks are online-optimised adaptations of traditional neuropsychological tests, while others instantiate novel designs. Extensive normative data are available on the tasks, encompassing detailed sociodemographic profiles to ensure fair assessment across diverse individuals[23,68,69]. Importantly,

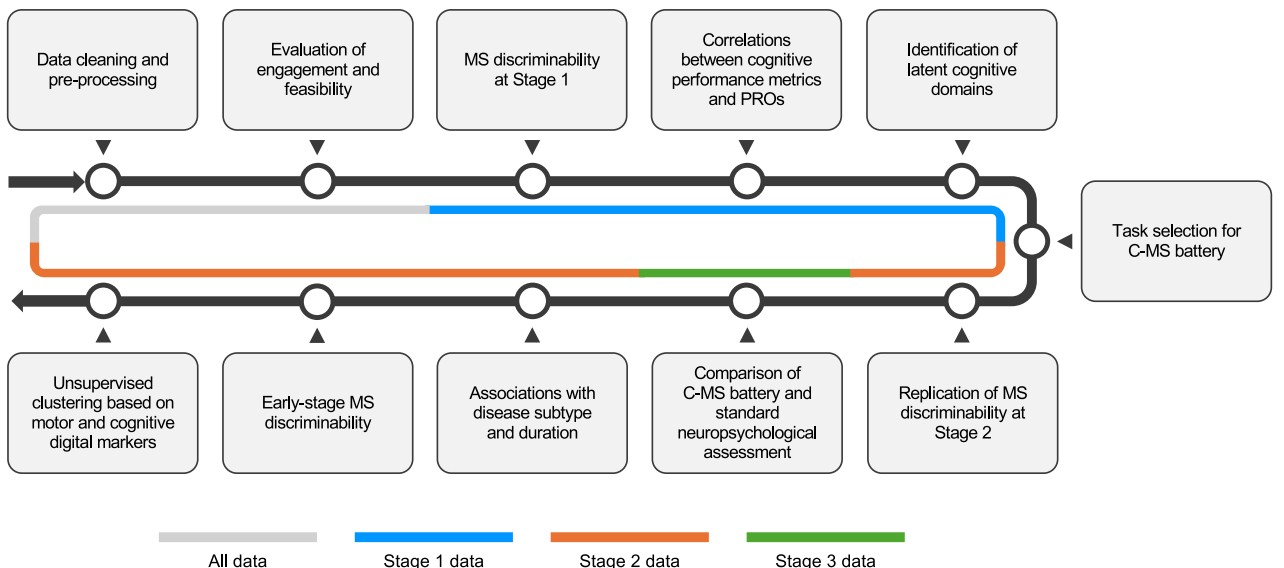

**Fig. 6 | Summary of statistical analyses.** Overview of the 11 main analysis steps performed in the study. The data used for each step are indicated by colour coding.

these tasks are deployable on any device with an internet connection (desktop computer, laptop, tablet, or smartphone), enhancing their usability and reach.

A superset of 22 tasks from the Cognitron library was selected for Stage 1 (Fig. 1A; task list and descriptions in Table S13). Selection criteria included scale of available normative data, alignment with other studies for cross-clinical group comparisons[22–24,27,29,70,71], and comprehensive coverage of different cognitive domains. To minimise testing time and fatigue effects while capturing data on all selected tasks, a stratified random sampling methodology was employed: each participant received a 12-task battery comprising 3 fixed tasks and 9 tasks randomly sampled from the remaining 19.

For Stages 2 and 3, participants were administered an identical 14-task battery in a fixed order, respectively remotely on their device of choice (Stage 2) or in person on a Samsung Galaxy ($N = 11$) or Apple ($N = 20$) tablet device (Stage 3). More cognitively demanding tasks were intentionally ordered earlier in the battery to mitigate fatigue effects. Twelve tasks were carried forward from Stage 1; two additional tasks were introduced in Stage 2 to augment the memory component of the battery. These newly introduced tasks were designed during the interim period between Stages 1 and 2 to target object memory and have superior sensitivity to subtle memory deficits compared to the word memory tasks[23,32]. Additionally, some task designs were modified between Stages 1 and 2 primarily to address changes in mobile browser compatibility, but also to expand problem space sampling; these changes generally did not have a significant impact on task performance (Fig. S15 and Supplementary Notes S3.1 and S3.2). In cases in which there was a significant association between design changes and task performance, normative data specific to the modified version of the task were utilised for norming (Supplementary Note S3.3). Spotter, an under-development version of the classic Digit Vigilance paradigm, underwent substantial design changes but lacked normative data on its modified version. As a result, it was excluded from analyses in Stages 2 and 3 data, pending future normative data availability. Nevertheless, data from this task will form the basis of future analyses investigating the relationship between fatigue and attention in MS.

Each task resulted in one accuracy-based and one response time-based score. Some tasks yielded additional scores, which typically relate to the cost in accuracy or response time across trials with differing levels of cognitive demand. For example, if the rule for response decisions switches between trials, or if some trials present distractors on screen while others do not (Table S13).

Normative data from the UK general population were collected as part of the Great British Intelligence Test (GBIT), a large-scale online longitudinal citizen science study conducted between December 2019 and February 2023 across 2 baseline and 4 follow-up timepoints[68,69,71]. For the present study, we used data from healthy individuals from the wider GBIT dataset, without neurological or psychiatric conditions, who provided complete sociodemographic information, were aged between 16 and 90 years, and were first-time task performers to ensure unbiased assessment ($N = 389,925$, $N$ by task in Table S14, sociodemographic characteristics in Table S15).

The standard neuropsychological assessment administered in Stage 3 included the following tests: California Verbal Learning Test–CVLT-II (Immediate and Delayed), Brief Visuospatial Memory Test–BVMT-R (Immediate and Delayed), Symbol Digit Modalities Test–SDMT, Trail Making Test–TMT, Stroop Test, Judgment of Line Orientation–JLO, Delis-Kaplan Executive Function System–D-KEFS Sorting Test, and Controlled Oral Word Association Test–COWAT. These tests were selected as they are commonly used in in-person clinical settings to comprehensively measure multidomain cognitive deficits occurring in MS[11,72,73]. In fact, 6 of the chosen tests overlap with the Minimal Neuropsychological Assessment of Cognitive Function in MS (MACFIMS), a comprehensive seven-test battery recommended for use in MS[74]. The Stroop Test and TMT were added because two of the tasks in the C-MS battery (Switching Stroop and Trail Making) are computerised adaptations of them. In-person testing procedures, test descriptions, and norming processes are detailed in Table S16 and Supplementary Note S4.

## Statistical analysis

Statistical analyses are detailed in the following sub-sections and visualised in Fig. 6. In general, extracted cognitive performance metrics and PRO scores were not all normally distributed; therefore, to ensure consistency and robustness, we applied non-parametric tests throughout. Specifically, two-way permutational ANOVA (PERMANOVA) was used to assess main effects when two factors were involved (e.g., disease subtype and duration), as it does not assume normality and handles unbalanced group sizes. Following significant main effects detected by PERMANOVA, post-hoc pairwise comparisons were conducted using the Mann–Whitney $U$ test, a non-parametric alternative to the $t$-test suitable for comparing independent groups without assuming normality. For analyses involving a single factor with more than two levels (e.g., cluster), the Kruskal–Wallis $H$ test–the non-

parametric analogue of one-way ANOVA–was used, followed by Dunn's test for post-hoc pairwise comparisons. An alpha threshold of 0.05 was applied to assess statistical significance throughout. In all post-hoc analyses, Bonferroni correction was applied to control for Type I error by adjusting the significance threshold based on the number of comparisons. For transparency, plots showing uncorrected pairwise results are provided in the Supplementary Information (Figs. S16–19).

**Data cleaning and preprocessing.** Participants <16 or >90 and those who provided invalid responses to, or did not complete, the questionnaire were excluded. Data from the Words/Objects Memory Delayed tasks were excluded if completed more than two hours after the Words/Objects Memory Immediate tasks, given the expected impact of time on forgetting. Summary performance metrics were derived for each task with records showing null values for accuracy and/or response time being discarded. Distributions of accuracies and response times in both normative and patient datasets were compared to establish thresholds for non-compliance and to eliminate outliers. Winsorisation and rank-inverse normal transformation were additionally evaluated as alternative outlier removal strategies to assess the robustness of findings and potential residual bias (Figs. S1 and S2). Using normative data, performance metrics were transformed into Deviation from Expected (DfE) scores, which are adjusted for sociodemographic factors (age, sex, dominant hand, education, ethnicity, first language, residence, occupation) and device type through regression-based normalisation[75] and expressed in standard deviation units. These scores indicate the level of cognitive impairment in pwMS compared to controls. As a sensitivity analysis, DfE scores were compared to standardised mean differences derived from propensity score-matched samples of pwMS and controls, matched on key sociodemographic covariates (Fig. S3). Summary scores for each PRO were obtained by summing scores on individual items. For the MSIS-29, only the motor subscale was evaluated. For the HADS, two summary scores were derived by summing scores on items related to anxiety (HADS-A) and depression (HADS-D) separately. Summary scores for MSIS-29 motor, MSWS-12, HADS-A and HADS-B were rescaled to a value in the range of 0–100 using a min-max normalisation procedure. For the MSWS-12, pwMS who were exempted from the questionnaire because they indicated an inability to walk, were included as scoring 100 in the normalised scale.

**Engagement and feasibility.** To assess feasibility, participation, completion, and return rates were evaluated for Stages 1 and 2. Participants' characteristics were compared across the following groups: (i) those who engaged with the assessment (completed at least one task) versus the overall UKMSR population, (ii) participants who completed the assessment versus those who did not, and (iii) participants who returned for Stage 2 after having taken part in Stage 1 versus those who did not. T-tests, chi-squared tests, and ANOVA were employed to evaluate these comparisons.

**Stage 1 - MS discriminability across task performance metrics and task selection for the C-MS battery.** In Stage 1, participants completed a randomly selected subset of 12 out of 22 tasks, with some failing to complete all assigned tasks. As a result, none of the participants completed every task, leading to a sparse dataset. Therefore, each task was analysed individually for its ability to differentiate between pwMS and controls, using all available data, rather than evaluating overall performance across tasks. Tasks for the C-MS battery were selected maximising MS discriminability while minimising device sensitivity and avoiding redundancy across latent cognitive domains as evaluated using factor analysis.

The median DfE (mDfE) score across participants for each performance metric served as an effect size indicator of task discriminability between pwMS and controls, employing Cohen's thresholds

(values of 0.2, 0.5 and 0.8 denote small, medium, and large effects, respectively). Correlations with PROs gauged performance metrics sensitivity to motor deficits, mental health, quality of life, fatigue, and disability severity. Kendall's Tau was selected as the main correlation metric due to the non-normal distribution of cognitive data. Correlations were calculated only using data for participants who completed both the relevant task and the corresponding PRO at one of the collection windows closest to the time of cognitive assessment.

Device sensitivity of each performance metric was evaluated using normative data to avoid disease-related confounders; specifically, eta squared values were derived from multiple linear regression models predicting performance metrics for each task and including as predictors sociodemographic variables (age, sex, dominant hand, education, ethnicity, first language, residence, occupation) and device type. In Cohen's notation, an eta squared value of 0.01 denotes a small effect, 0.06 indicates a medium effect, and values of 0.14 or higher represent large effects. Effects lower than 0.01 were considered negligible.

The latent cognitive structure underlying task performance was inferred by analysing factor loadings derived from factor analysis with varimax rotation, applied to a score matrix comprising the primary performance metric from each task. In the absence of complete data, and since values were missing completely at random for the tasks that had been randomly sampled, the pairwise deletion method was used to deal with missing values and produce the pairwise correlation matrix needed for the factor analysis. The optimal number of factors was determined using Kaiser's criterion (eigenvalues > 1), and varimax rotation was applied to enhance factor interpretability. To assess the validity of the factor solution, a validation procedure was conducted by iteratively splitting the dataset into two subgroups and independently performing factor analysis on each half (see Supplementary Note S1.1). Resulting factor loadings were compared to evaluate consistency across splits. This procedure was repeated across varying subgroup sample sizes to examine the stability of the factor structure at the available sample size (see Supplementary Note S1.2).

**Stage 2 - Replication of MS discriminability, associations with disease subtype and duration, and early-stage MS discriminability.** Replicability of MS discriminability across C-MS battery tasks was examined by correlating mDfE scores across performance metrics between Stage 1 and first-time Stage 2 participants (independent sample). The validity of the C-MS battery was evaluated by examining whether trends and associations observed in Stage 2 performance data aligned with findings from existing literature. To this end, we identified the cognitive domains most affected in MS, determined the prevalence of cognitive impairment within our sample, compared the characteristics of impaired and non-impaired groups, investigated associations between cognitive performance and disease subtype and duration, and evaluated early-stage MS discriminability. Participants were classified as cognitively impaired if they performed ≥1 SD (liberal), ≥1.5 SD (fair), or ≥2 SD (conservative) below matched controls on accuracy metrics or above on response time metrics, across ≥13% (liberal) or ≥20% (fair and conservative) of tested metrics, consistent with prior MS research[41,76]. These thresholds reflect varying levels of stringency and allow for comparison of impairment prevalence across different definitional criteria. Resulting impaired and non-impaired groups were compared by age, sex, disease duration, and subtype using t-tests and chi-squared tests. Associations with disease subtype, duration, and their interaction were tested using two-way permutational ANOVA. For performance metrics showing significant associations with these factors, Bonferroni-corrected post-hoc pairwise comparisons were conducted using the Mann–Whitney $U$ test. Early-stage MS discriminability was evaluated in participants

between 0 and 5 years post disease onset, looking at mDfE scores across accuracy and response time metrics and performing a series of Wilcoxon signed-rank tests against zero to evaluate significance.

**Stage 3 - Comparison of C-MS battery and standard neuropsychological assessment.** Comparative validity was assessed by determining the proportion of participants identified as impaired by the standard neuropsychological assessment who were also captured by the C-MS battery. Participants were classified as cognitively impaired on each battery if they scored ≥1.5 SD from matched controls on at least 20% of the measured parameters–a fair and commonly used threshold for detecting cognitive impairment[76]. For the C-MS battery, only primary performance metrics were retained. Furthermore, global cognitive composites, derived separately for each battery by averaging the retained performance scores, were correlated. An analogous approach was used to compare results from the Nine-Hole Peg Test and the Motor Control task. Participants were classified as impaired on these if they scored ≥1.5 SD from controls. It was assumed participants performed the Motor Control task with their less impaired hand. Consequently, only Nine-Hole Peg Test scores from the less impaired hand were retained.

**Unsupervised clustering of pwMS based on digital markers of motor and cognitive impairment derived via PROs and online cognitive assessment.** An alternative grouping of pwMS based solely on digital markers of impairment was derived using unsupervised clustering. Participants who completed both the Stage 2 cognitive assessment and the MSIS-29 motor and MSWS-12 PROs on the UKMSR during the Autumn 2022 collection window were included. *K*-means clustering with 4 clusters was selected as the primary clustering algorithm because it demonstrated the greatest validity and stability among the tested methods and cluster numbers (see Supplementary Note S1). Thirteen cognitive features (one primary performance metric expressed in DfE scores per task) alongside two motor features (one summary score per PRO) were used as clustering variables. These were all standardised by removing the mean and scaling to unit variance to ensure an equal contribution of all variables to the analysis and a fair comparison between variables of different types. Patterns in clustering variables across clusters were analysed to label the clusters. Additionally, patterns in other PROs as well as in sociodemographic and disease-related factors were examined to further characterise the clusters. Appropriate statistical tests—Kruskal–Wallis *H* tests, Dunn's post-hoc tests, ANOVA, and chi-squared tests—were applied to evaluate the significance of associations with cluster membership.

**Software.** Data preprocessing and statistical analysis were conducted in Python (3.11.7). The main libraries used were NumPy (1.26.4), pandas (2.1.4), statsmodels (0.14.0), SciPy (1.11.4), scikit-learn (1.2.2), factor-analyzer (0.5.1) and scikit-posthocs (0.9.0). Visualisations were created with matplotlib (3.8.0), seaborn (0.13.2), and PowerPoint.

### Reporting summary
Further information on research design is available in the Nature Portfolio Reporting Summary linked to this article.

## Data availability
All data in this study are maintained in a Trusted Research Environment (TRE) which is subject to rules related to access, security, and disclosability of data. All researchers requesting access to data must apply via, and be approved by, the UK MS Register Scientific Steering Committee (SSC), have completed the MRC GDPR training course, and be reputable researchers from a recognised academic institution. The SSC meets quarterly and (if approved) access to the data can be given within 30 days. Our participants' consent is given contingent upon data not being released from the TRE, due to the possibility of linking data from this dataset to extant datasets and the potential for it to become disclosive. For inquiries regarding data access, please contact Dr Rod Middleton at r.m.middleton@swansea.ac.uk. Source data are provided with this paper.

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

## Acknowledgements

We would like to thank all the people with Multiple Sclerosis who took part in the study across its three stages. This study would not have been possible without their contribution. A special thank you goes to the Brainstormers, who tested the assessment and offered their precious feedback during our Patient and Public Involvement (PPI) sessions prior to data collection. We also thank Dr. Marco Pitteri for his guidance on the norming of the neuropsychological tests included in Stage 3. A.L. conducted this work as part of the UK Research and Innovation Centre for Doctoral Training in Artificial Intelligence for Healthcare (http://ai4health.io, grant number EP/S023283/1), and as part of a postdoctoral position funded by the Institute of Psychiatry, Psychology & Neuroscience. A.M. and R.N. are supported by the Multiple Sclerosis Trials Collaboration (MSTC) and the Berkeley Foundation. The UK MS Register is funded by the UK MS Society (award 147). V.G. is supported by the Medical Research Council (grant number MR/W00710X/1). W.T. was funded by the EPSRC Centre for Doctoral Training in Neurotechnology at Imperial College London.

## Author contributions

Conceptualisation, study design and methodology–A.L., R.N., A.H., Supervision–R.N., A.H., Software design and support–A.H., P.J.H., W.T., Recruitment and support–R.M., K.T.D., Normative data collection–A.H., P.J.H., W.T., Stage 1 and 2 data collection–A.L., A.H., P.J.H., R.N., R.M., Stage 3 data collection–A.L., A.M., E.CARTA, R.N., K.T.D., R.M., Patient-report outcomes data collection–R.N., R.M., K.T.D., J.W., E.CRAIG, S.K., J.R., Normative data preprocessing–A.L., V.G., Stage 1 and 2 data preprocessing–A.L., Stage 3 data preprocessing–A.L., A.M., E.CARTA, Data analysis and visualisation–A.L., Writing original draft–A.L., Review and editing draft–A.L., A.M., V.G., E.CARTA, W.T., K.T.D., J.W., E.CRAIG, S.K., J.R., E.COCCO, P.J.H., R.M., R.N., A.H.

## Competing interests

A.H. is the founder and director of Future Cognition Ltd and co-founder and co-director of H2CD, which develops custom online assessment software and provides online assessment technology as a service for third parties, primarily in the research and healthcare sectors. P.J.H. is co-founder and co-director of H2CD. W.T. works as an employee of H2CD. R.N. has carried out paid advisory board and research trial with Roche and Novartis. He is vice chair of the National Institute for Health and Care Excellence HTA committee C. E.CO has received honoraria for consultancy or speaking from Biogen, Novartis, Sanofi, Merck, Roche, Bristol, Janssen, and Alexion. All other authors report no competing interests.
