## [Transparent Peer Review file · Nature Communications]

Large-scale online assessment uncovers a distinct Multiple Sclerosis subtype with selective cognitive impairment

Corresponding Author: Dr Annalaura Lerede

Version 0:

Reviewer comments:

Reviewer #1

(Remarks to the Author)
Dear Authors,

I enjoyed reading your manuscript examining a remotely-administered cognitive assessment battery in pwMS. Overall, I think your manuscript is quite solid and comprehensive. I have a few suggestions that I think will help make your strong manuscript clearer and stronger:

MAJOR:

- Based on your participant flow diagram, it appears as though you have a bias in your participants. Individuals who were not registered in the online portal were not invited to participate in all stages and only participated in stage 3. People who choose not to register themselves may inherently have different traits/characteristics/skills than the ones who register or have less proficiency with use of a computer/accessing the internet. This needs to be discussed.
- The manuscript is very, very long. Perhaps so long that the research community will not read and appreciate your findings. It might benefit your work to move more information into supplementary files to streamline the main manuscript.
- Your abstract mentions the identification of a prevalent cognitive phenotype in pwMS, but the discussion about this phenotype is quite limited (i.e., you do not further characterize this phenotype or discuss your phenotype amongst the growing literature of cognitive phenotypes in MS). Given that your title suggests this is the crux of your manuscript, I think it should be discussed differently or in more depth. I hesitate to say that since your manuscript is already too long, but the lack of this information is antithetical to your title. Perhaps reframe your title to reflect the purposes listed at the end of your introduction instead of a cognitive phenotype.

MINOR:

- With full knowledge that the remote assessment literature is moving in this direction, I do think we still have to be mindful of elements in the participant environment that cannot be controlled like they would in an in-person environment. Including a comment about potential variability due to multiple screen sizes, distractions, etc. might be helpful
- I do not see any discussion about fatigue as it relates to study demands. You asked for significant time and mental energy, the study demands on participants were likely quite fatiguing.
- It looks as though Cognitron has not yet been used in pwMS, this should be mentioned

Reviewer #2

(Remarks to the Author)
Summary

This study demonstrates the feasibility and validity of large-scale online cognitive assessment for people with Multiple Sclerosis (pwMS), using data from the UK MS Register (UKMSR). The goal was to propose and validate an approach to expand the sample size and range of cognitive tasks beyond those used in most MS studies involving cognition. Using an established platform for cognitive assessment (Cognitron), the authors derived Cognitron-MS (C-MS), a 12-task cognitive

battery optimized for detecting MS-related impairments, and validated its performance against standard neuropsychological assessments. They conducted this work in three stages, with Stage 1 evaluating 22 tasks using a random task sampling approach to derive the 12-task C-MS battery, Stage 2 validating C-MS on an independent sample (with some overlap) of pwMS by comparison to Stage 1 outcomes, and Stage 3 comparing C-MS with a standard in-person neuropsychological assessment,

Key Findings

1. Feasibility: High participation and completion rates were observed for Stages 1 and 2, demonstrating the accessibility and usability of the online assessment approach. Question: Can the authors provide some insight for why only 41.3% of Stage 1 participants returned for Stage 2?
2. Validation: The C-MS battery showed high discriminative ability and strong correlations to the full Cognitron battery (Phase 1) in assessing cognitive impairments across multiple domains. The trends observed in Stage 2, such as the cognitive domains showing the highest discriminability, prevalence of global cognitive impairment, and severity of impairment across subtypes, aligned with the MS literature. C-MS also compared well to in-person neuropsychological assessment, but the sample size (31) was much smaller than the other two stages.
3. Clustering Analysis: Agglomerative hierarchical clustering revealed four distinct subgroups of pwMS, varying in motor and cognitive impairments, with significant implications for personalized treatment and diagnosis.
4. Disease Associations: Cognitive impairment prevalence correlated with age, disease duration, and progressive MS subtypes, and early-stage deficits were identified with significant sensitivity.

Data, Methodology and Validity

The data interpretation and conclusions are overall robust and supported by detailed analyses. However, several areas require clarification or further revision:

1. Clustering Methodology: In the section of "unsupervised clustering of pwMS based on digital markers of motor and cognitive impairment derived via PROs and online cognitive assessment", agglomerative hierarchical clustering was effectively employed. However, some critical details are missing, such as the specific distance metric and linkage method used. Can the authors provide these details and explain their choice of distance metric and linkage method? Additionally, how was the stability and validity of the identified clusters confirmed?
2. Factor Analysis: In the section of "Stage 1 - MS discriminability across task performance metrics and task selection for C-MS battery", factor analysis was used to identify latent cognitive domains, and the screen plot in Fig. S1 provides partial clarification but lacks a discussion of the explained variance and criteria for factor retention, and details on the factor extraction criteria and validation (e.g., cross-validation) are missing. Can the authors elaborate on these details?
3. Clinical Validity: In the section of "Stage 3 - Comparison of C-MS battery and standard neuropsychological assessment", the thresholds for defining cognitive impairment (e.g., 1.5 SD) are mentioned, but the rationale behind these cutoffs is unclear. Can the authors elaborate on why these thresholds were selected and their clinical relevance? Also, the standard neuropsychological assessment detected 16 out of 31 pwMS, while C-MS detected 20, with only 12 overlapping. Can the authors comment on this discrepancy?
4. Outlier Removal and Normalization: In the section of "Data cleaning and preprocessing", while outlier thresholds and regression-based normalization are described, there is no mention of sensitivity analyses to confirm that these preprocessing steps do not bias results. Can the author add more details?
5. PRO Linkage: In the section of "Engagement and feasibility and unsupervised clustering of pwMS based on digital markers of motor and cognitive impairment", the timing of PRO data collection relative to cognitive assessments (e.g., six-month windows) introduces potential temporal confounds. Further justification or sensitivity analyses would be helpful here.
6. Task-Specific Adjustments: The exclusion of certain tasks (e.g., Spotter) due to lack of normative data requires a clearer explanation of its impact on the overall analysis.
7. Multiple Comparison Corrections: in the section of "Statistical analysis", Bonferroni corrections are mentioned but not detailed in terms of how they were applied. Can the authors clarify the methodology for applying multiple comparison corrections and their effect on significance thresholds? How were the Bonferroni corrections implemented, and what effect did they have on the adjusted significance thresholds?
8. Statistical Details: in the section of "Statistical analysis", can the authors expand on the specific non-parametric methods used and why they were chosen over parametric tests for certain datasets? It is also helpful to provide additional rationale for using 2-way permutational ANOVA and post-hoc pairwise comparisons, including handling of type I errors.

Clarity and Context

The manuscript provides sufficient context for the study, situating it within the broader field of MS research.

Presentation of results

The figures are clear and well-designed, but some figures (e.g., Fig. 1A) have low resolution which make them difficult to read.

References

The manuscript appropriately references foundational studies.

Overall Impression

The quality of the data, robustness of the analyses, and potential significance of the findings are all strengths of this study. While the study is methodologically sound and presents novel insights, addressing the above issues would enhance its clarity, transparency, and overall impact.

Reviewer #3

(Remarks to the Author)

Version 1:

Reviewer comments:

Reviewer #1

(Remarks to the Author)

Dear Authors,

I appreciate the thoughtful comments to not only my suggestions but those of the other reviewer. In my opinion, you have now sufficiently addressed the concerns and I have no additional concerns that would necessitate further revision.

Reviewer #2

(Remarks to the Author)

The authors have provided a very thorough response, and have addressed all my concerns in their revisions. I (and my ECR co-reviewer) thank the authors for responding with such care and attention.

Reviewer #3

(Remarks to the Author)

Manuscript title: Large-Scale online assessment uncovers a distinct Multiple Sclerosis subtype with selective cognitive impairment

Manuscript ID: NCOMMS-24-72661-T

Corresponding author: Dr Annalaura Lerede

Date: 28-05-25

REVIEWER COMMENTS

We thank the Editor and Reviewers for reading our manuscript and providing constructive comments and suggestions. We have made substantial changes based on these comments and feel that they have improved the quality and clarity of the manuscript. Below, we provide a detailed point-by-point response to each comment and describe the revisions made accordingly.

Reviewer #1 (Remarks to the Author):

Dear Authors,

I enjoyed reading your manuscript examining a remotely-administered cognitive assessment battery in pwMS. Overall, I think your manuscript is quite solid and comprehensive. I have a few suggestions that I think will help make your strong manuscript clearer and stronger:

Thank you for your positive and helpful comments. We have addressed all your comments in the main text and feel that our article is now strengthened.

MAJOR:

- Based on your participant flow diagram, it appears as though you have a bias in your participants. Individuals who were not registered in the online portal were not invited to participate in all stages and only participated in stage 3. People who choose not to register themselves may inherently have different traits/characteristics/skills than the ones who register or have less proficiency with use of a computer/accessing the internet. This needs to be discussed.

Bias, engagement and digital poverty are important considerations. We agree that those who do not engage in online technology are likely, on average, to have different characteristics and have reported on this in other populations. Conversely, the use of a digital register cohort does provide a larger and more inclusive sample of people with MS than other study approaches to date that have been conducted in this population. We now provide a more balanced discussion of the inclusiveness and representativeness of the sample as follows, including consideration of the limitations.

“Although the use of an established digital registry enabled the cognitive assessment of a large and inclusive MS sample, a limitation is that individuals unfamiliar with or less willing to engage in online technologies may not have taken part in Stages 1 and 2, introducing selection biases. More broadly, digital poverty remains a known challenge for such technologies. As per work in

other populations¹, future studies should characterise the relationship between engagement in online assessments and demographics in MS, and explore ways to mitigate these barriers, but also consider applying the same technology under supervised conditions for the minority of pwMS where this is necessary².”

- The manuscript is very, very long. Perhaps so long that the research community will not read and appreciate your findings. It might benefit your work to move more information into supplementary files to streamline the main manuscript.

Thank you - communicating detailed analyses like these succinctly is important and challenging. We have worked through the article end-to-end to further streamline the text and improve readability, without compromising essential content. Additionally, we have added a brief signpost at the beginning of each results paragraph to clearly indicate its purpose and relevance, and to help the reader not to get lost in the results and remain engaged. Our article is now around 4000 words, which is well within the journal's limit of 5000 words (not including Abstract, Methods, References and Figure legends).

- Your abstract mentions the identification of a prevalent cognitive phenotype in pwMS, but the discussion about this phenotype is quite limited (i.e., you do not further characterize this phenotype or discuss your phenotype amongst the growing literature of cognitive phenotypes in MS). Given that your title suggests this is the crux of your manuscript, I think it should be discussed differently or in more depth. I hesitate to say that since your manuscript is already too long, but the lack of this information is antithetical to your title. Perhaps reframe your title to reflect the purposes listed at the end of your introduction instead of a cognitive phenotype.

Thank you for this insightful comment. We agree that the identification and characterisation of cognitive phenotypes in MS and for that matter their relationship to underlying mechanisms is important and that there is insufficient space to cover these topics in detail in our manuscript. It is well established that cognitive problems in MS are common and there has been substantial work trying to identify which aspects of cognition are most vulnerable. However, what we show here is that a substantial proportion of MS patients can have selective cognitive impairments, that is, in the absence of the motor deficits that are the focus of today's clinical evaluation. Our determining that this is the case in a large cohort has been enabled by our technology approach to assessment. We now make this point clear by doing the following in the Discussion section:

- Extending the discussion on the characteristics of this selective cognitive phenotype.
- Highlighting the importance of additionally running cognitive assessment alongside motor assessment.
- Briefly discuss how underlying disease mechanisms could be driving selective cognitive impairment.
- Note the need for biomarker analyses to adjudicate possible mechanisms underlying selective cognitive impairment.

We have also modified the closing statement of our Introduction as follows:

“Finally, we applied data-driven clustering to cognitive performance data from the C-MS battery and motor disability data from patient-reported outcomes (PROs) to identify and characterise distinct symptom-based phenotypes, including a common subtype with selective cognitive impairment. In doing so, we aimed to advance individualised symptom profiling in MS by incorporating detailed cognitive characterisation, which is often underrepresented in large-scale MS studies due to the lack of appropriate and scalable assessment tools.”

Finally, we have modified our title as follows:

“Large-scale online assessment uncovers a distinct Multiple Sclerosis subtype with selective cognitive impairment”

MINOR:

- With full knowledge that the remote assessment literature is moving in this direction, I do think we still have to be mindful of elements in the participant environment that cannot be controlled like they would in an in-person environment. Including a comment about potential variability due to multiple screen sizes, distractions, etc. might be helpful

Conducting assessments online poses specific methodological challenges. Here, we used an established platform (Cognitron) with tasks specifically designed to minimise sensitivity to differences in the devices that participants own and are tested on. As shown in Fig. S7, most tasks deployed in this study are not sensitive to differences in participants' devices. Nevertheless, we included device type as a covariate in all regression-based models, particularly to control for residual variability in the response time metrics, which are more susceptible to differences in participants' devices.

Study protocols included instructions to find time and a quiet space, avoid distractions, and try to complete in one session. Data preprocessing included automated checks to detect interruptions, window switching, and indicators of poor task compliance. The Cognitron platform deploys the assessment uniformly and scores responses automatically, reducing the potential for experimenter or scoring bias. The platform has proven useful for deploying assessments across a range of neurological populations.

Looking ahead, we are currently developing performance modelling software that can differentiate the contributions of visuomotor and cognitive deficits to performance, the application of which will form the basis of future studies in MS. We now discuss this more explicitly in the Introduction and in the Discussion sections.

- I do not see any discussion about fatigue as it relates to study demands. You asked for significant time and mental energy, the study demands on participants were likely quite fatiguing.

We agree that fatigue is a highly relevant factor in MS. To mitigate fatigue effects during testing we have done the following:

- In Stage 1, task assignment and order were randomised across participants, averaging out potential fatigue-related effects.
- In Stage 2, we intentionally ordered more cognitively demanding tasks earlier in the session, with simpler tasks placed later.
- Participants were encouraged to take small breaks between tasks as needed.

Additionally, we examined the correlation between cognitive performance and self-reported fatigue using the Fatigue Severity Scale (FSS), but, consistent with previous studies³, correlations were largely negligible or non-significant (Fig. 2C). Another indirect indicator of fatigue is whether participants completed the full assessment, which we report on the paper. Despite the study's demands, high completion rates were observed—largely within a single session—comparable to those in general population studies, suggesting good engagement and tolerance of the assessment load.

In our ongoing research, we also include a sustained attention task with embedded fatigue analogue scales (Digit Vigilance Task) to specifically and objectively measure this domain. However, this task was still under development during Stage 1, and although included at the end of the battery in Stage 2, we did not analyse it due to the lack of normative data. However, these data will form the basis of a separate detailed analysis focused on the relationship between attention and fatigue in people with MS.

We now clarify these points in the Methods and Discussion sections and will continue to expand on objective measures of fatigue in future work.

- It looks as though Cognitron has not yet been used in pwMS, this should be mentioned.

We did pilot Cognitron in people with MS from the UK Register (the Brainstormers PPI group) prior to the deployment in the study reported here. That work was not published. Additionally, people with MS have taken part in our past citizen science and epidemiological studies—for example, several hundred people with MS participated in the Great British Intelligence Test⁴. Therefore, for precision, we now state as follows at the end of the Introduction section.

“This was the first large-scale deployment of the Cognitron assessment platform for people with MS.”

Reviewer #2 (Remarks to the Author):

Summary

This study demonstrates the feasibility and validity of large-scale online cognitive assessment for people with Multiple Sclerosis (pwMS), using data from the UK MS Register (UKMSR). The goal was to propose and validate an approach to expand the sample size and range of cognitive tasks beyond those used in most MS studies involving cognition. Using an established platform for cognitive assessment (Cognitron), the authors derived Cognitron-MS (C-MS), a 12-task cognitive battery optimized for detecting MS-related impairments and validated its performance against standard neuropsychological assessments. They conducted this work in three stages, with Stage 1 evaluating 22 tasks using a random task sampling approach to derive the 12-task C-MS battery, Stage 2 validating C-MS on an independent sample (with some overlap) of pwMS by comparison to Stage 1 outcomes, and Stage 3 comparing C-MS with a standard in-person neuropsychological assessment,

Key Findings

1. Feasibility: High participation and completion rates were observed for Stages 1 and 2, demonstrating the accessibility and usability of the online assessment approach. Question: Can the authors provide some insight for why only 41.3% of Stage 1 participants returned for Stage 2?

It is a known characteristic of digital health registries that participants tend to engage in some but not all timepoints. Which participants engage at each timepoint varies, so each timepoint includes a subset of the 'active' registry members. The nature of the data is therefore very large but inherently sparse. We have previously published methods for handling such large-sparse data in the context of patient-reported outcomes (PROs) collected via the UK MS Register⁵. We acknowledge that the return rate for cognitive testing in Stage 2 (41.3%) was lower than that observed for the established PRO measures collected through the same platform over the same period (e.g. MSIS-29: 71.1%, MSWS-12: 65.7%, HADS: 72.7%). However, a lower return rate for the cognitive testing was expected at this stage for several reasons:

- Participants were explicitly informed at the start of both Stages 1 and 2 that the cognitive assessment tool was still under development.
- In Stage 2, we further specified that individual results would not be provided, which may have reduced participant motivation to engage with the assessment.
- The cognitive assessment required considerably more time and effort compared to brief questionnaire-based PROs.
- The PROs are well-established/expected by this MS cohort.

We anticipate improved return rates in future iterations of the platform as the cognitive battery undergoes further optimisation, is formally validated across independent MS cohorts, and is repositioned from a developmental prototype to a standardised, clinically relevant digital assessment tool. Increased visibility following the publication of results from this and related

studies, alongside the integration of personalised feedback for participants are also expected to enhance participant engagement and longitudinal adherence.

Importantly, it was promising that individuals who were more likely to have a higher level of impairment did not have a lower rate of return across the two timepoints. We now make these points explicitly in the Discussion section as follows:

“Another positive outcome was that 41.3% of Stage 1 participants returned for Stage 2, despite the cognitive battery being under development, more demanding than PROs, and not yet returning individual results. Returnees were on average older and had a higher prevalence of progressive MS, suggesting that those at greater risk of cognitive impairment were not less likely to re-engage. While this return rate was lower than that observed for PROs (e.g., MSIS-29: 71.1%), it is expected to improve as the battery becomes more optimised, widely validated, and familiar to participants.”

2. Validation: The C-MS battery showed high discriminative ability and strong correlations to the full Cognitron battery (Phase 1) in assessing cognitive impairments across multiple domains. The trends observed in Stage 2, such as the cognitive domains showing the highest discriminability, prevalence of global cognitive impairment, and severity of impairment across subtypes, aligned with the MS literature. C-MS also compared well to in-person neuropsychological assessment, but the sample size (31) was much smaller than the other two stages.

Thank you, this is an accurate summary.

3. Clustering Analysis: Agglomerative hierarchical clustering revealed four distinct subgroups of pwMS, varying in motor and cognitive impairments, with significant implications for personalized treatment and diagnosis.

Thank you, this is an accurate summary.

4. Disease Associations: Cognitive impairment prevalence correlated with age, disease duration, and progressive MS subtypes, and early-stage deficits were identified with significant sensitivity.

Yes, this is correct.

Data, Methodology and Validity

The data interpretation and conclusions are overall robust and supported by detailed analyses. However, several areas require clarification or further revision:

1. Clustering Methodology: In the section of "unsupervised clustering of pwMS based on digital markers of motor and cognitive impairment derived via PROs and online cognitive assessment", agglomerative hierarchical clustering was effectively employed. However, some critical details are missing, such as the specific distance metric and linkage method used. Can the authors provide

these details and explain their choice of distance metric and linkage method? Additionally, how was the stability and validity of the identified clusters confirmed?

Thank you, this is a really good point. To exhaustively answer it, we evaluated the stability of our clustering solution given our sample size (N=1180) and compared it with:

- using the same clustering algorithm (Agglomerative Hierarchical Clustering) but different linkage methods
- using a different clustering algorithm (K-means)
- using a different number of clusters (we tested one on either side, so 3 and 5 clusters)
- using principal component analysis to reduce the number of clustering features before applying the chosen clustering algorithm

Details on how the stability analysis was implemented and results are now presented in Supplementary Note 2.

In brief, the stability analysis revealed improved stability when using K-means with 4 clusters over Agglomerative Hierarchical Clustering with any number of clusters. Therefore, we decided to use K-mean as the main clustering algorithm and we updated the main manuscript accordingly. While two of the clusters have slightly changed, the main conclusions have not:

- there is a cluster of people with MS mainly showing cognitive deficits; the magnitude of the average deficits (mDfE scores) for this cluster is now larger than in the previous solution and the deficits are more widespread across cognitive domains;
- cluster membership is associated with age, gender, disease subtype, disease duration, and education but not with age at onset;
- cluster composition is heterogeneous, and there is a partial dissociation between motor and cognitive deficits;
- there is a higher prevalence of males in the cluster with severe motor and cognitive deficits;
- webEDSS reflects motor more than cognitive severity.

Details on the updated clustering algorithm are now included in the Methods section. Figure 5 and the corresponding Results and Discussion sections have been updated accordingly.

Overall, we believe that these additional analyses greatly strengthen confidence in the stability of the main results of the data-driven analyses.

2. Factor Analysis: In the section of "Stage 1 - MS discriminability across task performance metrics and task selection for C-MS battery", factor analysis was used to identify latent cognitive domains, and the screen plot in Fig. S1 provides partial clarification but lacks a discussion of the explained variance and criteria for factor retention, and details on the factor extraction criteria and validation (e.g., cross-validation) are missing. Can the authors elaborate on these details?

We have now reworked our reporting of the factor analysis to improve clarity and completeness. As now specified in the Results section, the number of factors retained was based on the Kaiser's criterion (eigenvalues >1). The eigenvalues, the variance explained by each factor (before and after varimax rotation), and the scree plot are reported in Supplementary Table S6 and Fig. S4. The factor analysis was conducted directly on the pairwise correlation matrix, which was

evaluated using the pairwise deletion method to handle missing scores. This was appropriate given that task assignment in Stage 1 was random and consequently missingness was completely random. Overall, we retained 6 factors (because they had eigenvalues > 1), which explained 28.6% of the total variance.

To assess the validity and stability of the factor structure, we implemented two additional analyses, now detailed in Supplementary Note 1:

- Split-half validation: the dataset was repeatedly split into two subgroups, and factor analysis was independently performed on each half. Factor loadings were matched across subgroups, and average correlations were computed over 100 iterations. We observed strong consistency across splits (average matched-factor correlation = 0.74), with qualitatively similar loading patterns (example in Fig. S5).
- Sample size stability: to assess whether stability had been reached at our available sample size, we repeated the above procedure while varying the sample size from 100 to 1524 (in increments of 20). The resulting stability curve (Fig. S6) was reaching a plateau at the maximum sample size tested, indicating that the 6-factor solution is robust at the available sample size.

While there is some residual variability in factor structure, this likely reflects inherent noise in cognitive task performance and the sparsity of the score matrix. Nonetheless, a matched-factor correlation of 0.74 was deemed sufficient for our intended purpose—namely, to inform the selection of non-redundant tasks.

We have updated the relevant Methods and Results sections of the main text to reflect these changes.

3. Clinical Validity: In the section of "Stage 3 - Comparison of C-MS battery and standard neuropsychological assessment", the thresholds for defining cognitive impairment (e.g., 1.5 SD) are mentioned, but the rationale behind these cutoffs is unclear. Can the authors elaborate on why these thresholds were selected and their clinical relevance? Also, the standard neuropsychological assessment detected 16 out of 31 pwMS, while C-MS detected 20, with only 12 overlapping. Can the authors comment on this discrepancy?

Thank you for this suggestion. As discussed in the literature^{6,7}, prevalence estimates of cognitive impairment in MS vary considerably depending, among other factors, on the threshold used to define impairment. In our manuscript, we adopted a criterion of 1.5SD below (for accuracy) and above (for response time) the normative mean on at least 20% of the evaluated test parameters. This is considered a fair and balanced threshold and is among the most used in MS research, including in several key studies⁸. However, the choice of threshold is ultimately arbitrary and depends on the specific purpose of the assessment—for instance, more liberal thresholds are preferred when the goal is to detect early impairment—limiting comparability across studies⁶.

To illustrate how prevalence estimates vary with different levels of stringency in our dataset (as anticipated in the literature), we also applied both stricter and more liberal thresholds (based on Fischer et al. 2014⁶) in Stage 2 analyses. We aimed to highlight the inherent arbitrariness of threshold selection and underscore the advantages of digital cognitive assessments. These tools yield continuous performance metrics and detailed behavioural logs, enabling more granular quantification of cognition. They also allow for large-scale, repeated assessments—including at

baseline—overall supporting a shift from binary impairment classifications to a more dynamic, function-based approach, as increasingly advocated in the field⁹.

The discrepancy between the two assessments likely reflects differences in design and sensitivity. The C-MS battery includes tasks specifically designed to offer greater precision than standard neuropsychological scales, offers broad coverage of cognitive domains, and benefits from higher trial numbers, richer task performance metrics, and a larger UK-based normative dataset that allows for robust sociodemographic adjustments. These features can increase sensitivity to subtle impairments that may be missed by traditional scales. Similar patterns have been already observed in previous studies developing and validating Cognitron-based batteries across various neurological populations^{2,10,11}. As for the 4 participants identified as impaired only by the standard neuropsychological assessment, it is worth noting even traditional neuropsychological tests show modest test-retest reliability and incomplete overlap when impairment is defined categorically. Some degree of discrepancy is therefore expected, especially near threshold cutoffs.

We have reworked our text in the Method and Discussion sections to touch on these topics more clearly.

4. Outlier Removal and Normalization: In the section of "Data cleaning and preprocessing", while outlier thresholds and regression-based normalization are described, there is no mention of sensitivity analyses to confirm that these preprocessing steps do not bias results. Can the author add more details?

Thank you for raising this important point. Previous work investigating the effects of data cleaning and preprocessing steps in the same Cognitron normative dataset found that regardless of whether extensive or liberal cleaning methods were applied, little to no effect was found in the distribution of sociodemographic variables, and known associations between sociodemographics and cognitive performance metrics were preserved¹². Nevertheless, to further assess the robustness of our findings and validate the preprocessing pipeline in the context of the present study, we conducted the following sensitivity analyses:

1. We evaluated the robustness of our results to different outlier removal strategies by reprocessing the data using three alternative methods and comparing the resulting median Deviation from Expected (mDfE) scores across tasks with the original ones. The mDfE scores were computed using regression-based normalisation as in the main analysis. The alternative outlier removal methods were:
 - a. Winsorisation at $\pm 3SD$ using mean and SD computed from the combined sample (patient and control groups)
 - b. Winsorisation at $\pm 3SD$ using mean and SD computed from the control group only
 - c. Rank-inverse normal transformation (applied to the combined sample)

Results are shown in Fig. S1 (accuracies) and Fig. S2 (response times). Across all methods, mDfE scores were highly correlated with those from the original method (all Pearson's $r > 0.94$, $p < 0.0001$), for both accuracy and response time metrics. This indicates that the choice of outlier removal method did not substantially impact nor bias

the results. As also shown in previous work¹², the large sample size often available when working with Cognitron likely reduces the influence of individual outliers on group-level statistics.

2. As a sensitivity analysis for the regression-based normalisation approach used to derive DfE scores, we performed a parallel analysis using propensity score matching between patient and control data based on the same sociodemographics included in regression-based normalisation and evaluated the standardised mean difference (SMD) between the matched scores. Then we compared these resulting scores with the mDfE scores obtained from regression-based normalisation across tasks. This is the same method that we have previously applied for other of our online datasets with similarly favourable results that support our primary DfE approach (e.g., Hampshire et al. 2024¹³). Specifically:
 - a. For each task, we included only participants with available data on that task.
 - b. We aggregated patient and control data and trained a binomial logistic regression model to estimate each participant's propensity score (i.e., probability of being a patient given sociodemographic characteristics).
 - c. Propensity scores were binned into equal-width bins (initially 10). Within each bin, we identified the smaller group size (patients or controls) and randomly sampled that number from both groups to retain balanced pairs. This process was repeated, increasing the number of bins, until the standard deviation of propensity scores between matched groups was below 0.1.
 - d. After matching, we computed SMD in task scores between the matched patients and control, along with 95% confidence intervals derived using a normal approximation based on the standard error of the SMD.
 - e. We compared the resulting SMD values with the original mDfE scores across tasks for both accuracy and response time metrics.

Results, presented in Fig. S3, revealed very high correlations between the two methods (accuracy: $\text{corr}=0.90$, $p<0.0001$; response time: $\text{corr}=0.97$, $p<0.0001$), indicating that the use of regression-based normalisation is valid while having the advantage of enabling a more comprehensive analysis of relevant factors while retaining all as opposed to partial data.

In summary, the performed sensitivity analyses demonstrated that our results are robust to different outlier removal approaches and normalisation procedures. We have now added corresponding statements in the main manuscript to reflect this.

5. PRO Linkage: In the section of "Engagement and feasibility and unsupervised clustering of pwMS based on digital markers of motor and cognitive impairment", the timing of PRO data collection relative to cognitive assessments (e.g., six-month windows) introduces potential temporal confounds. Further justification or sensitivity analyses would be helpful here.

We thank the reviewer for this helpful comment. We will try to specify better the timing of patient-reported outcome (PRO) data collection relative to cognitive assessments and hopefully provide more clarity.

Since 2018, PROs have been collected every six months (in spring and in autumn) in the UK MS Register, with each collection window lasting 28 days and extended, when necessary, through reminder emails to maximise response rates.

For Stage 2, data collection was conducted between **November 8, 2022, and January 13, 2023**, with 90.6% of the data being collected in the first month. These data were matched to PRO data collected during the autumn 2022 window (**September 13 – October 24**), which was temporally closer to the cognitive assessment than the subsequent spring 2023 window (**March 1 – April 12**). Although concurrent collection would have been ideal, the UK MS Register opted to schedule cognitive testing immediately after the autumn PRO collection window to avoid interfering with established PRO collection procedures and engagement patterns. Selecting the earlier PRO window also ensured consistency in the ordering of assessments, with PROs preceding cognitive testing.

For Stage 1, data collection was more spread out (**October 22, 2021 – May 15, 2022**), with 88.8% of the data being collected during January and February 2022. This overlapped closely with both the autumn 2021 PRO window (**September 7 – October 14**) and spring 2022 windows (**March 9 – April 20**). In this case, each participant's cognitive data was matched to the temporally closest available PRO dataset, with priority given to minimising time gaps rather than enforcing a consistent assessment order.

We have now clarified this temporal matching procedure in the manuscript and acknowledge the potential for temporal confounding. However, we believe this approach is reasonable given the limited control we had over the scheduling.

6. Task-Specific Adjustments: The exclusion of certain tasks (e.g., Spotter) due to lack of normative data requires a clearer explanation of its impact on the overall analysis.

Spotter was newly introduced to Cognitron as a web-optimised variant on the classic Digit Vigilance Task at the time of Stage 1. It was still undergoing active development and improvement when it was included in the Stage 1 MS battery. The most up-to-date version of this task, which was deployed in Stage 2, lacked normative data at the time of Stage 2 data collection and analysis. Nonetheless, we included the task in the battery to facilitate future validation, based on promising preliminary results from Stage 1 and its relevance to domains commonly affected in people with MS, specifically, sustained attention and fatigue. Importantly, Spotter was presented last in the battery for all participants, and as such, its exclusion has no impact on the results presented here, having not been analysed with the rest of the data and not included in any of the primary or secondary outcomes.

The Spotter task variants and their optimisation in MS form the basis of a separate detailed analysis focused on how to measure the relationship between attention and fatigue, which is the focus of another paper currently in preparation. We judged that the inclusion of this task development work would overcomplicate the current paper, although the results are promising. We have clarified this point in the main text as follows:

“Spotter, an under-development version of the classic Digit Vigilance paradigm, underwent substantial design changes but lacked normative data on its modified version. As a result, it was excluded from analyses in Stages 2 and 3 data, pending future normative data availability. Nevertheless, data from this task will form the basis of future analyses investigating the relationship between attention and fatigue in MS.”

7. Multiple Comparison Corrections: in the section of "Statistical analysis", Bonferroni corrections are mentioned but not detailed in terms of how they were applied. Can the authors clarify the methodology for applying multiple comparison corrections and their effect on significance thresholds? How were the Bonferroni corrections implemented, and what effect did they have on the adjusted significance thresholds?

Thank you, we agree that additional detail was needed, and we have now clarified the methodology for multiple comparison corrections in the revised manuscript.

Bonferroni corrections were applied following a significant main effect when post-hoc pairwise comparisons were conducted to identify which groups differed. In these cases, the Bonferroni correction was implemented by dividing the pre-comparison significance level for each test by the number of pairwise comparisons to be made. This was necessary in three cases:

1. **Disease duration:** when doing pairwise comparisons between disease duration groups for tasks showing a significant main effect of duration.
2. **Disease subtype:** when doing pairwise comparisons between disease subtypes for tasks showing a significant main effect of subtype.
3. **Data-driven clusters:** when doing pairwise comparisons between clusters for both cognitive and motor features, showing a significant main effect of cluster membership.

Applying the Bonferroni correction resulted in more conservative significance thresholds, and in all three contexts, this led to fewer significant pairwise differences compared to the uncorrected analyses. For transparency, we now include plots of uncorrected pairwise results in the Supplementary Information for comparison.

We have also updated the Statistical Analysis section of the Methods to more clearly describe how and when Bonferroni corrections were applied.

“An alpha threshold of 0.05 was applied to assess statistical significance throughout. In all post-hoc analyses, Bonferroni correction was applied to control for Type I error by adjusting the significance threshold based on the number of comparisons. For transparency, plots showing uncorrected pairwise results are provided in the Supplementary Information (Fig. S16-19).”

8. Statistical Details: in the section of "Statistical analysis", can the authors expand on the specific non-parametric methods used and why they were chosen over parametric tests for certain datasets? It is also helpful to provide additional rationale for using 2-way permutational ANOVA and post-hoc pairwise comparisons, including handling of type I errors.

Thank you, we have now expanded the Methods section accordingly and combined it with the previous statement for clarity and conciseness.

“In general, extracted cognitive performance metrics and PRO scores were not all normally distributed; therefore, to ensure consistency and robustness, we applied non-parametric tests throughout. Specifically, two-way permutational ANOVA (PERMANOVA) was used to assess main effects when two factors were involved (e.g., disease subtype and duration), as it does not assume normality and handles unbalanced group sizes. Following significant main effects detected by PERMANOVA, post-hoc pairwise comparisons were conducted using the Mann-Whitney U test, a non-parametric alternative to the t-test suitable for comparing independent groups without assuming normality. For analyses involving a single factor with more than two levels (e.g., cluster), the Kruskal-Wallis H test—the non-parametric analogue of one-way ANOVA—was used, followed by Dunn’s test for post-hoc pairwise comparisons. An alpha threshold of 0.05 was applied to assess statistical significance throughout. In all post-hoc analyses, Bonferroni correction was applied to control for Type I error by adjusting the significance threshold based on the number of comparisons. For transparency, plots showing uncorrected pairwise results are provided in the Supplementary Information (Fig. S16-19).”

Clarity and Context

The manuscript provides sufficient context for the study, situating it within the broader field of MS research.

Presentation of results

The figures are clear and well-designed, but some figures (e.g., Fig. 1A) have low resolution which make them difficult to read.

Figure 1A has been updated and should now have high resolution.

References

The manuscript appropriately references foundational studies.

Thank you.

Overall Impression

The quality of the data, robustness of the analyses, and potential significance of the findings are all strengths of this study. While the study is methodologically sound and presents novel insights, addressing the above issues would enhance its clarity, transparency, and overall impact.

Thank you for these helpful comments. We have addressed all of them and agree that this has further strengthened our article.

Reviewer #3 (Remarks to the Author):

Thank you.

REFERENCES

1. Cai, Z. *et al.* Online46: online cognitive assessments in elderly cohorts - the British 1946 birth cohort case study. Preprint at <https://doi.org/10.1101/2024.09.19.24313984> (2024).
2. Del Giovane, M. *et al.* Computerised cognitive assessment in patients with traumatic brain injury: an observational study of feasibility and sensitivity relative to established clinical scales. *EClinicalMedicine* 59, 101980 (2023).
3. Johnson, S. K., Lange, G., DeLuca, J., Korn, L. R. & Natelson, B. The Effects of Fatigue on Neuropsychological Performance in Patients With Chronic Fatigue Syndrome, Multiple Sclerosis, and Depression. *Appl Neuropsychol* 4, 145–153 (1997).
4. Hampshire, A., Ballard, C. & Williams, G. Computerized neuropsychological tests undertaken on digital platforms are cost effective, achieve high engagement, distinguish and are highly sensitive to longitudinal change: Data from the PROTECT and GBIT studies. *Alzheimer's & Dementia* 16, e041122 (2020).
5. Lerede, A. *et al.* Patient-reported outcomes in multiple sclerosis: a prospective registry cohort study. *Brain Commun* 5, fcad199 (2023).
6. Fischer, M. *et al.* How reliable is the classification of cognitive impairment across different criteria in early and late stages of multiple sclerosis? *J Neurol Sci* 343, 91–99 (2014).
7. Amato, M. P. *et al.* Cognitive assessment in multiple sclerosis—an Italian consensus. *Neurological Sciences* 39, 1317–1324 (2018).
8. Benedict, R. H. B. Standards for sample composition and impairment classification in neuropsychological studies of multiple sclerosis. *Multiple Sclerosis Journal* 15, 777–778 (2009).
9. Pitteri, M., Ziccardi, S., Dapor, C., Guandalini, M. & Calabrese, M. Lost in classification: Lower cognitive functioning in apparently cognitive normal newly diagnosed RRMS patients. *Brain Sci* 9, 321 (2019).
10. Bălăeț, M. *et al.* Online cognitive monitoring technology for people with Parkinson's disease and REM sleep behavioural disorder. *NPJ Digit Med* 7, 118 (2024).
11. Shibata, K. *et al.* Remote digital cognitive assessment reveals cognitive deficits related to hippocampal atrophy in autoimmune limbic encephalitis: a cross-sectional validation study. *EClinicalMedicine* 69, 102437 (2024).
12. Trender, W. Validating the Cognitron online cognitive assessment technology for large scale remote citizen science, clinical trials and individual assessment applications. (Imperial College London, London, 2025).
13. Hampshire, A. *et al.* Cognition and Memory after Covid-19 in a Large Community Sample. *New England Journal of Medicine* 390, 806–818 (2024).